# Topology and geometry of the learning space of ReLU networks: connectivity and singularities

**Marco Nurisso**[1*]**, Pierrick Leroy**[1*]**, Giovanni Petri**[2] **& Francesco Vaccarino**[1]
[1]Department of Mathematical Sciences, Politecnico di Torino, Turin, Italy
[2]NPLab, Network Science Institute, Northeastern University London, London, UK

## ABSTRACT

Understanding the properties of the parameter space in feed-forward ReLU networks is critical for effectively analyzing and guiding training dynamics. After initialization, training under gradient flow decisively restricts the parameter space to an algebraic variety that emerges from the homogeneous nature of the ReLU activation function. In this study, we examine two key challenges associated with feed-forward ReLU networks built on general directed acyclic graph (DAG) architectures: the (dis)connectedness of the parameter space and the existence of singularities within it. We extend previous results by providing a thorough characterization of connectedness, highlighting the roles of bottleneck nodes and balance conditions associated with specific subsets of the network. Our findings clearly demonstrate that singularities are intricately connected to the topology of the underlying DAG and its induced sub-networks. We discuss the reachability of these singularities and establish a principled connection with differentiable pruning. We validate our theory with simple numerical experiments.

## 1 INTRODUCTION

The success of deep learning has spurred extensive research into the geometry and dynamics of neural network training. While classical results primarily focus on layered architectures, many modern networks adopt more flexible structures, such as directed acyclic graphs (DAGs), arising either from design or from pruning and compression strategies. These architectures challenge existing theory and necessitate new tools to understand their training behavior, particularly in the presence of non-smooth, homogeneous activations like ReLU.

In this paper, we study two fundamental training pathologies in DAG-based ReLU networks: the *(dis)connectedness* of the training-invariant parameter space, and the presence of *singularities* within it. Our analysis is grounded in the observation that training with gradient flow on networks with homogeneous activations gives rise to symmetry-induced conservation laws. These laws constrain learning trajectories to an algebraic variety—referred to as the *invariant set*—defined by a system of quadratic equations dependent on the network's topology and initialization.

Our contributions are as follows:

- We derive an elegant formulation for the conservation laws that arise during gradient flow training of ReLU networks as a result of rescaling symmetries, in the general case of DAG-based architectures.

- We extend previous results on shallow networks (Nurisso et al., 2024) by studying the geometry and topology of the invariant set in the general architecture case, providing necessary and sufficient conditions for its connectedness based on network bottlenecks and balance constraints.

---

*Equal contribution. Corresponding authors `{marco.nurisso,pierrick.leroy}@polito.it`

- We identify and analyze singularities of the invariant set, showing that they correspond to disconnected sub-networks, and prove that they are unreachable under standard gradient flow from generic initializations.

- We propose a nuclear norm-based regularizer that promotes convergence to singular configurations, thereby enabling differentiable, structure-agnostic pruning. In our experiments, we observe that L1 regularization—despite not explicitly targeting neuron sparsity—empirically induces similar singular behavior as our dedicated regularizer, and therefore also fosters effective lossless pruning.

Taken together, our results shed light on the interplay between network topology and optimization geometry. They also offer a principled pathway for designing pruning mechanisms that exploit the structure of the optimization space rather than relying solely on heuristic sparsity constraints.

## 1.1 RELATED WORKS

**DAG neural networks.** General feedforward architectures can be formalized as directed acyclic graphs (DAGs) (Gori et al., 2023; Hwang & Tung, 2023; Chirag Agarwal et al., 2021), or more abstractly as quivers(Armenta & Jodoin, 2021), though this perspective remains relatively under-explored. DAG structures support variants of topological sorting (Kahn, 1962), which recover the notion of layers (Boccato et al., 2024a; Chirag Agarwal et al., 2021). Both natural and synthetic neural systems align well with this broader formalism (Boccato et al., 2024b; Milano et al., 2023). DAG-like networks can also emerge through unstructured pruning (see below) or via the sampling of sparse subnetworks, as in the lottery ticket hypothesis (Frankle & Carbin, 2018; Liu et al., 2018; You et al., 2019).

**Pruning.** Pruning methods are typically classified as structured or unstructured (Hoefler et al., 2021; Cheng et al., 2024). Structured pruning targets entire groups of parameters, such as neurons or channels (Yuan & Lin, 2006; Nonnenmacher et al., 2021), while unstructured pruning removes individual weights (Han et al., 2015; Frantar & Alistarh, 2023), often at the cost of hardware inefficiency. Sparsity can be induced iteratively during training (Lin et al., 2020; Jin et al., 2016), or through differentiable techniques and regularization (Savarese et al., 2020; Pan et al., 2016; Kang & Han, 2020). Recent efforts like *any-structural pruning* Fang et al. (2023) aim to unify pruning strategies into general frameworks. Our work approaches pruning through the geometry of the optimization landscape: a nuclear norm regularizer naturally promotes sparsity across arbitrary structures in DAG-based networks, though current computational limitations restrict its practical use.

**Singularities and deep learning.** Singular Learning Theory (Watanabe, 2009; 2007) blends statistical learning with algebraic geometry, treating singularities as central to the learning process in the Bayesian framework, when working with non-identifiable models such as neural networks. Recent works have applied its tools to describe modern neural network architectures (Wei et al., 2022; Lau et al., 2023; Furman & Lau, 2024). Singularities are also foundational to neuro-algebraic geometry (Marchetti et al., 2025), which examines the space of functions realizable by networks—often termed the *neuro-manifold*. The influence of singularities on learning has been recognized for decades: early work (Amari & Ozeki, 2001; Amari et al., 2001; 2006) analyzed their impact on gradient descent in simplified settings. Their relation to network topology has also been studied; for instance, skip connections are known to reduce singularities (Orhan & Pitkow, 2017).

**Training dynamics of ReLU networks.** A large body of work analyzes the gradient flow and descent dynamics of networks with homogeneous activations, including convergence guarantees (Sirignano & Spiliopoulos, 2020; Mei et al., 2018; Rotskoff & Vanden-Eijnden, 2022) and implicit bias properties (Boursier et al., 2022; Chizat & Bach, 2020; Lyu et al., 2021; Soudry et al., 2018). ReLU's non-smoothness poses analytic challenges (Eberle et al., 2021), yet its positive homogeneity enables rescaling symmetries (Dinh et al., 2017) and conservation laws (Neyshabur et al., 2015; Marcotte et al., 2023; Kunin et al., 2020; Zhao et al., 2022; Tanaka et al., 2020; Nurisso et al., 2024). The discrete-time setting of gradient descent has also received attention (Feng et al., 2019; Smith et al., 2021; Kunin et al., 2020). More broadly, symmetry principles continue to shed light on deep learning phenomena (Grigsby et al., 2023; Głuch & Urbanke, 2021; Bronstein et al., 2021; Ziyin et al., 2025).

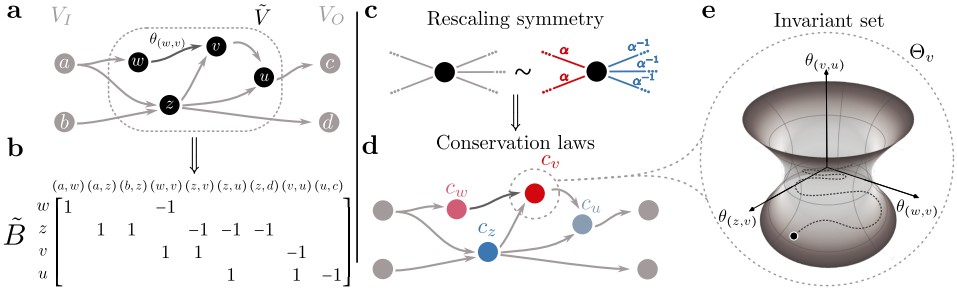

Figure 1: **a.** Example of a feed-forward DAG architecture $G$. **b.** The incidence matrix $\tilde{B}$ of $G$ with rows associated to input and output neurons removed. **c.** Visualization of the rescaling symmetry of ReLU neurons. **d.** The initialization determines the balance value $c_v = \langle\!\langle \theta, \theta \rangle\!\rangle_v$ of every hidden neuron, which characterizes the shape of the invariant set (**e**).

## 2 SETUP AND NOTATION

In this section, we start by introducing our notation for the neural network topology. We then review known results on symmetries of ReLU networks and their associated conservation laws, reformulating them in our compact notation. Next, we introduce the notion of *invariant set*, whose properties are further investigated in section 3: first its connectedness, and then its singularities, each part including numerical experiments. We conclude and discuss limitations in section 4.

**DAG neural networks.** Consider a general computational graph $G$ describing a feed-forward neural network architecture (see the related work section 1.1). $G$ is a directed, acyclic graph (DAG) on a set of nodes $V$, called neurons, with edges $E$. We identify a subset of neurons $V_I \subseteq V$ containing the *input neurons*, such that no edges are entering the elements of $V_I$, and a subset $V_O \subseteq V$ of *output neurons* such that no edges are going out of the elements in $V_O$. We assume that $V_I \cap V_O = \varnothing$, i.e., no neurons have empty neighbors. We write $\partial V$ to denote the set of input and output neurons $V_I \sqcup V_O$. All the other nodes $\tilde{V} \subseteq V$ are the *hidden nodes* which are to be the fundamental computational unit of the neural network $V = \tilde{V} \sqcup \partial V$ (Figure 1a). For any node $v \in V$, we call $\mathrm{Anc}(v)$ the set of its *ancestors*, i.e. nodes $w \in V$ such that there exists a path in $G$ from $w$ to $v$, and $\mathrm{Desc}(v)$ its *descendants*, i.e. the nodes $w \in V$ such that there exists a path $v \to v_1 \to \cdots \to v_n \to w$ in $G$.

Each edge $(i, j) \in E$ has a parameter $\theta_{(i,j)} \in \mathbb{R}$ associated with it and, when data is passed through the network, hidden nodes $v \in \tilde{V}$ sum the values of their incoming edges, apply the ReLU function $\sigma$ and output to each outgoing edge the resulting value multiplied by the edge parameter. As it is standard in the literature (see e.g. 5.1 in Bishop & Nasrabadi (2006)), one can also consider biases in this setup by adding a "virtual" input neuron whose input is fixed to 1 and adding edges from it to every hidden neuron.

We call *parameter space* $\Theta$ the set of all parameters, i.e. the vector space of real functions over the edges $\theta : E \to \mathbb{R}$, $\Theta \cong \mathbb{R}^{|E|}$ and we write $f_G(\cdot; \theta)$ to indicate the input-output function encoded by $G$ with parameters $\theta$. Throughout the paper, it will be convenient to formulate the results by describing the connectivity structure with the incidence matrix $B$ of $G$ (Bondy & Murty, 1979). $B \in \mathbb{R}^{|V| \times |E|}$ is a standard object in graph theory that describes how each edge is connected to its endpoints. Its elements are defined as follows: $B_{v,(i,j)} = 1$ if $v = j$, $B_{v,(i,j)} = -1$ if $v = i$ and 0 otherwise. See, for example, the DAG in Figure 1a and its associated incidence matrix (with rows associated to nodes in $\partial V$ removed) in Figure 1b.

**Symmetries of ReLU networks.** In this work, following Du et al. (2018), we study the properties of neural network where the activation function $\sigma$ is *homogeneous*, namely $\sigma(x) = \sigma'(x) \cdot x$ for every $x$ and for every element of the sub-differential $\sigma'(x)$ if $\sigma$ is non-differentiable at $x$. The commonly used ReLU ($\sigma(z) = \max\{z, 0\}$) and Leaky ReLU ($\sigma(z) = \max\{z, \gamma z\}$ with $0 \le \gamma \le 1$) activation functions satisfy this property.

It is well known that the geometry of the parameter space $\Theta$ is heavily influenced by the properties of the activation function. Some activation functions and specific neural network modules induce

some symmetries in the parameter space, i.e., transformations $h$ of the parameters that do not change the function encoded by the network $f_G(\cdot; \theta) = f_G(\cdot; h \circ \theta)$ (Zhao et al., 2025). In the case of homogeneous activations, the most critical symmetry is given by rescaling (Neyshabur et al., 2015). In fact, the input weights of any hidden neuron can be rescaled by a positive scalar $\alpha > 0$, provided that its output weights are rescaled by the inverse $\alpha^{-1}$. This result is well-known for single and multi-layer networks and holds even in general DAG architectures as it is defined at a single node. We write this as the action of the group $\mathbb{R}_+$ of positive real numbers on each local parameter space $\Theta_v := \{\theta_{(x,y)} \mid (x,y) \in E \text{ and } x = v \text{ or } y = v\}$ by means of $T_\alpha^v(\theta) = T_\alpha^v((\theta_{(i,v)})_i, (\theta_{(v,j)})_j) = ((\alpha\theta_{(i,v)})_i, (\frac{1}{\alpha}\theta_{(v,j)})_j)$ (Figure 1c).

**Local conservation laws under gradient flow.** The presence of symmetries in the neural network's parameter-function map induces the presence of same-loss sets of the loss landscape. Let indeed $f_G(\cdot; \theta) : \mathbb{R}^d \to \mathbb{R}^e$, $D = \{(x_i, y_i) \in \mathbb{R}^d \times \mathbb{R}^e\}_{i=1}^N$ be a training dataset and $L : \Theta \to \mathbb{R}$ be a loss function which depends on the parameters only through the output of the neural network[1], that is

$$L(\theta) = \frac{1}{N} \sum_{i=1}^N \ell(f_G(x_i; \theta), y_i) \tag{1}$$

where $\ell : \mathbb{R}^{|V_O|} \times \mathbb{R}^{|V_O|} \to \mathbb{R}$ is differentiable.

Let us now assume that we train the network using the continuous-time analog of gradient descent i.e. *gradient flow* (GF), $\dot{\theta}(t) \in -\nabla_\theta L(\theta(t)) := -g(\theta(t))$, where $\nabla_\theta L(\theta(t))$ is the Clarke sub-differential (Clarke et al., 2008). Given that the loss function $L$ depends on the parameters only through $f_G$, its value at $\theta$ must be constant over the orbit of rescaling. This, together with the fact that the gradient of a differentiable function at a point is orthogonal to the level set at that point, means that the gradient is orthogonal to the orbit under the action of rescaling, at any parameter $\theta$ where $L(\theta)$ is differentiable. This orthogonality condition constrains the possible values of the gradient and, by extension, the possible gradient flow trajectories.

This orthogonality condition can be shown to be

$$\langle\!\langle \theta, g(\theta) \rangle\!\rangle_v := \sum_{i: i \to v} \theta_{(i,v)} g(i,v) - \sum_{j: v \to j} \theta_{(v,j)} g(v,j) = 0$$

for every hidden neuron $v \in \tilde{V}$ (see Section A.2 for details): the gradient modulated by the parameter values is a *network flow*, as the quantity $g(\theta) \odot \theta$ is conserved when passing through each hidden neuron, where $\odot$ denotes the element-wise Hadamard product between vectors. This result is well known and is obtained, with a different approach, in e.g. Tanaka et al. (2020).

We propose to conveniently re-write the gradient conditions at all hidden nodes using a variation of the incidence matrix $B$ of $G$.

**Proposition 1.** *Let $\tilde{B} \in \mathbb{R}^{|\tilde{V}| \times |E|}$ be the incidence matrix of $G$ with the rows associated with input and output nodes removed; then $\langle\!\langle \theta, g(\theta) \rangle\!\rangle_v = 0 \; \forall v \in \tilde{V}$ is equivalent to*

$$\tilde{B}(\theta \odot g(\theta)) = 0. \tag{2}$$

*Proof.* The proof follows directly from the definition of the incidence matrix. At any hidden node $v \in \tilde{V}$, if we denote by $\theta_e$ the weight of any $e \in E$, and by $g_e$ the $e$-th component of $g(\theta)$, then it holds

$$\tilde{B}(\theta \odot g(\theta))_v = \sum_{e \in E} B_{v,e} \theta_e g_e = \sum_{i: i \to v} \theta_{(i,v)} g_{(i,v)} - \sum_{j: v \to j} \theta_{(v,j)} g_{(v,j)} = \langle\!\langle \theta, g(\theta) \rangle\!\rangle_v = 0.$$

$\square$

**Invariant sets.** Equation (2), implies that some quantities are conserved under gradient flow optimization or, equivalently, that the learning trajectories are constrained to a lower-dimensional subset of the parameter space.

---

[1]This means that we do not include regularization terms which depend explicitly on the parameters.

**Proposition 2.** *Let $f_G$ be initialized with parameters $\theta(0)$ such that $\tilde{B}\theta(0)^2 = c \in \mathbb{R}^{|\tilde{V}|}$, with $\theta(0)^2$ the element-wise square of the vector $\theta(0)$, then, for every $t \geq 0$, it holds that $\tilde{B}\theta(t)^2 = c$.*

*Proof.*

$$\frac{\mathrm{d}}{\mathrm{d}t}\tilde{B}\theta(t)^2 = \tilde{B}\frac{\mathrm{d}}{\mathrm{d}t}\theta(t)^2 = 2\tilde{B}(\theta(t) \odot \dot{\theta}(t)) = -2\tilde{B}(\theta(t) \odot g(\theta(t))) = 0.$$

$\square$

This result (visualized in Figure 1d), note, is a general, elegant re-writing of the well-known neuron-wise conservation law (Du et al., 2018; Liang et al., 2019; Kunin et al., 2020; Saxe et al., 2013). As we will see, this is not a mere notational feature, as this formulation reveals precious insights into the relationship between the training dynamics and the neural network's graph structure. We now define the *invariant set* as the set the training trajectories are constrained to due to the conservation laws: if the network is initialized in the invariant set, it will remain in it until the end of training.

**Definition 1** (Invariant set, generalization of Nurisso et al. (2024))**.** *Given $c = (c_v)_{v \in \tilde{V}}$, we call* invariant set *the set $\mathcal{H}_G(c) \subseteq \Theta$ of the solutions of the system of polynomial equations $\tilde{B}\theta^2 = c$.*

If we look at the single equation associated with hidden neuron $v \in \tilde{V}$, we see that $\tilde{B}\theta^2 = c$ can be written as

$$\sum_{i \to v} \theta_{(i,v)}^2 - \sum_{j \leftarrow v} \theta_{(v,j)}^2 = c_v \tag{3}$$

which corresponds to a hyperbolic *quadric hypersurface* in the local parameter space of $v$, $\Theta_v$ (Figure 1e). From the graph's point of view, we can interpret this as stating that the vector of squared parameters $\theta^2$ is akin to a fluid flowing through the edges of $G$, with input/output nodes acting as unconstrained sources/sinks and hidden nodes supplying or demanding some flow according to the value and sign of $c$ (Ford & Fulkerson, 1962).

In Nurisso et al. (2024), it is shown that for shallow networks, the geometrical structure of $\mathcal{H}_G(c)$ is simple, as the total invariant set factorizes into the Cartesian product of the neurons' quadric hypersurfaces. In the general case (MLP or DAG), the situation is much more complex because the equations of $\mathcal{H}_G(c)$ are *coupled*: parameters associated with internal edges appear in multiple equations.

The invariant set $\mathcal{H}_G(c)$, which is an *algebraic variety* (albeit we do not know whether it is reduced or not), is an interesting object because it lies in between the redundant but more "concrete" parameter space and the abstract function space (or *neuromanifold* (Calin, 2020; Kohn, 2024)) the model's implemented function lives in. In fact, fixed any $c$, no two parameters in $\mathcal{H}_G(c)$ are observationally equivalent w.r.t. rescalings, that is $f_G(\cdot; \theta) = f_G(\cdot; (T_\alpha^v(\theta_v))_v)$ for no rescaling, thus making the invariant set a good proxy for the function space $\mathcal{F}_G := \{f_G(\cdot; \theta) \,|\, \theta \in \Theta\}$. Nevertheless, as discussed in Nurisso et al. (2024), different values of $c$ correspond to different topologies of $\mathcal{H}_G(c)$, meaning that $\mathcal{F}_G$ does not provide the full picture to understand the learning process. $\mathcal{H}_G(c)$ also has its limits: it might not be identifiable with the functions it contains. Indeed, for two isomorphic nodes $i, j \in G$ with $i \neq j$, permuting their input and output weights will yield an observationally equivalent parametrization. And if $i$ and $j$ are such that $c_i = c_j$, then both parameters will be in $\mathcal{H}_G(c)$, and so the parameterization map $\theta \mapsto f_G(\cdot; \theta)$ will not be injective into $\mathcal{F}_G$.

## 3 GEOMETRY AND TOPOLOGY OF THE INVARIANT SET

The study of the geometric and topological properties of $\mathcal{H}_G(c)$ (Definition 1) can give us interesting loss and data-independent insights into the training processes.

$\mathcal{H}_G(c)$ is the set of solutions of a system of degree-two polynomial equations, each one corresponding to a quadric. Despite the apparent simplicity, studying general intersections of quadrics is not easy (de Medrano, 2023) but, in our case, the specific structure of $\mathcal{H}_G(c)$ greatly simplifies the process. In fact, the equations $\tilde{B}\theta^2 = c$ correspond to a system of *coaxial* quadric hypersurfaces, meaning that they contain only squares and no mixed terms of the form $\theta_e \theta_{e'}$. This fact allows us to employ some powerful recent results in topology (de Medrano, 2023).

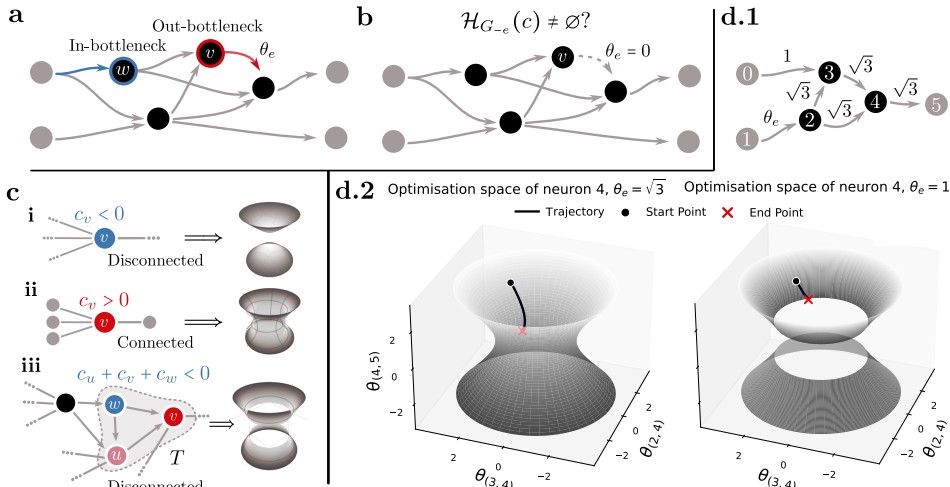

Figure 2: **Overview of connectedness. a.** In- and out-bottlenecks in $G$. **b.** The non-emptiness of $\mathcal{H}_G(c)$ is guaranteed if every hidden neuron has input and output edges. **c.** Different connectedness conditions and intuitive visualizations of the associated algebraic varieties for an out-bottleneck **d.** Numerical experiment showcasing training dynamics in a connected and disconnected scenario for a DAG network with 3 hidden nodes (**d.**1).

### 3.1 NON-EMPTINESS

The first result concerns the non-emptiness of $\mathcal{H}_G(c)$ for a given $c$ or, from the other point of view, what the possible balance values $c$ that can appear from an initialization are.

**Proposition 3** (Feasible balance). *For all $c \in \mathbb{R}^{|\tilde{V}|}$, one has $\mathcal{H}_G(c) \neq \varnothing$.*

This result (proven in Section A.3) means that any balance configuration on the hidden neurons $c$ is achievable through a parameter initialization, provided that no neurons are excluded from the computations. From the point of view of network flows, this tells us that it is possible to build a flow $\theta^2$ that satisfies any supply and demand on the hidden nodes.

### 3.2 CONNECTEDNESS

Nurisso et al. (2024) showed that, for some values of $c$, the invariant set of a shallow ReLU neural network is disconnected. This means that a network initialized in one connected component cannot reach an optimum located in another through gradient flow. Here, we show that the conditions for connectedness in the general DAG case resemble the ones for the shallow case, with additional pathological cases resulting from particular graph topologies.

To find out whether $\mathcal{H}_G(c)$ is connected or not, we will use the following proposition, adapted to our case from de Medrano (2023).

**Proposition 4** (de Medrano (2023) Proposition 4.7). *$\mathcal{H}_G(c)$ is connected if and only if, for every $e \in E$, $\mathcal{H}_{G_{-e}}(c) \neq \varnothing$, where $G_{-e} = (V, E \smallsetminus \{e\})$.*

In other words, $\mathcal{H}_G(c)$ is connected if it is "robust enough" so that the deletion of single edges does not change the possibility of satisfying the supply and demand conditions of $c$. From this observation, together with Proposition 3, we see that the cases in which there is disconnection must necessarily come from the presence of neurons with only one input or output connection.

**Definition 2** (Bottleneck neurons). *A hidden neuron $v \in \tilde{V}$ is an in-bottleneck if $\deg^-(v) = 1$ and an out-bottleneck if $\deg^+(v) = 1$. We denote with $V_B^-, V_B^+$ the sets of in and out-bottleneck nodes, respectively. For any out-bottleneck neuron $v \in V_B^+$, we call $\overline{\mathrm{Anc}}(v)$ the set of its pure ancestors, i.e. ancestors $w \in \mathrm{Anc}(v)$ such that any path from $w$ to $V_O$ passes through $v$. Analogously, any in-bottleneck defines a set of pure descendants $\overline{\mathrm{Desc}}(v)$ containing neurons $w \in \mathrm{Desc}(v)$ such that any path from $V_I$ to $w$ passes through $v$. Among the pure ancestors of an out-bottleneck $v$, we say*

*a set $T \subset \overline{\mathrm{Anc}}(v)$ is* closed by forward edges *if the inclusion $\bigcup_{u \in T \setminus \{v\}} \mathcal{N}^+(u) \subset T$ holds, where $\mathcal{N}^+(u)$ denotes the out-neighbors of $u$. The analogous notion of closedness by backward edges is obtained by considering descendants and in-neighbors instead.*

The removal of the single connection of a bottleneck neuron (Figure 2a,b) will disconnect it, and the set of its pure ancestors/descendants will be effectively cut out from the network's computation: either because they receive no inputs from $V_I$ or because they produce no output to $V_O$.

It turns out that it is possible to derive a complete characterization of connectedness and disconnectedness, leveraging tools from network flow theory.

**Theorem 1.** *$\mathcal{H}_G(c)$ is connected if and only if $\forall v \in V_B^+, \forall T \subset \overline{\mathrm{Anc}}(v)$ s.t. $T$ closed by forward edges $\sum_{u \in T} c_u \geq 0$ and $\forall v \in V_B^-, \forall T \subset \overline{\mathrm{Desc}}(v)$ s.t. $T$ closed by backward edges $\sum_{u \in T} c_u \leq 0$.*

*Proof.* The proof is fairly technical and can be found in Section A.4. □

Intuitively, disconnectedness is caused by bottleneck nodes such that cutting their single edge makes the balances (supplies/demands) of their pure ancestors/descendants unfeasible. For instance, let us look at the out-bottleneck $v$ in Figure 2c: (i) $c_v < 0$ means that $v$ requires more flow coming out than in. This is not feasible because there are no other output connections. (ii) $c_v > 0$ means that $v$ requires more flow coming in than out, which is always feasible for the small shallow network (ii). More generally, this is feasible in bigger networks unless (iii) there is a forward closed set $T$ for which $\sum_{k \in T} c_k < 0$, as predicted by Theorem 1. Intuitively, there is too much flow coming in $T$ than can be absorbed before $v$. Mechanistically, there is a disconnection whenever the output/input weight of a bottleneck neuron cannot change sign through gradient flow.

Two immediate corollaries follow, clarifying the problem of connectedness in most practical cases.

**Corollary 1.** *If $G$ has no bottleneck neurons, then $\mathcal{H}_G(c)$ is connected.*

Note that bottleneck neurons are rare in MLPs because their existence implies the presence of a layer with only one node. The exceptions are given by neural networks built to solve binary classification or scalar regression problems, where there is a single output neuron, and the neurons in the last layer are all out-bottlenecks. In that case, we adapt Theorem 1, finding that the results of Nurisso et al. (2024) nicely carry over to the multi-layer case.

**Corollary 2.** *If $G$ is a fully-connected MLP such that all hidden layers contain more than one neuron, then $\mathcal{H}_G(c)$ is connected if and only if $c_v \geq 0 \ \forall v \in V_B^+$ and $c_v \leq 0 \ \forall v \in V_B^-$.*

*Proof.* The result follows from Theorem 1 by noticing that in a fully connected architecture, the only pure ancestor/pure descendant of a node is the node itself. □

A practical implication of this section is that the expressivity of ReLU networks can be reduced at initialization to the extent that they lose their universal approximation capability. That is, some functions become immediately unreachable, regardless of the chosen loss function or dataset.

**Numerical experiments** We illustrate Theorem 1 on the toy DAG neural network shown in Figure 2d.1, implemented using dedicated software (Boccato et al., 2024a) and in the gradient descent setting. Neuron 4 will be the out-bottleneck of interest. All hidden neurons (in black) have ReLU activation. With the initialization given by the values in the figure, we have $c = (c_2, c_3, c_4) = (\theta_e^2 - 6, 1, 3)$. There are 3 forward closed sets of nodes, the largest being $T = \{2, 3, 4\}$ with $\sum_{k \in T} c_k = \sum_k c_k = \theta_e^2 - 4$. Therefore, if at initialization $\theta_e(0) < \sqrt{2}$, then $\sum_{k \in T} c_k < 0$ and the optimization space will disconnect at neuron 4. Concretely, it means that the balance condition at neuron 4 will forbid sign switches of $\theta_{(4,5)}$.

We try to make the model learn the function $f : (x_1, x_2) \to -(x_1 + x_2)$ for positive inputs. The only way to output negative values is if $\theta_{(4,5)} < 0$, and so the optimum will not be reachable for $\theta_e(0) = 1 < \sqrt{2}$ as shown on the right plot of Figure 2d.2, while the optimum is reachable (and is reached) for $\theta_e(0) = \sqrt{3} > \sqrt{2}$, as depicted on the left plot. Additional experiments can be found in Section A.9.1.

Figure 3: **a.** A singularity of the invariant set corresponds to a configuration where a set of neurons is cut out from input and outputs. **b.** Visualization of the training dynamics on an invariant set with singularities. **c.** Proportion of null singular values along training for shallow network with 20 hidden neurons with and without regularization. **d.** Test losses as a function of the number of neurons pruned. Shaded regions in **c.** and **d.** denote confidence intervals over 50 independent trainings that have converged to a low loss solution.

## 3.3 SINGULARITIES

While singularities in the neuromanifold—i.e., the function space—have been extensively studied (Amari & Ozeki, 2001; Amari et al., 2001; 2006; Henry et al., 2024), in this part we propose a complementary perspective by analyzing singularities of the invariant set $\mathcal{H}_G(c)$, which sits in the optimization (parameter) space.

**Singularities are sub-networks.** In algebraic geometry, a *singularity* refers to a point in a variety where the tangent space is not well-defined. Mathematically, if a variety $X$ is given by the system of $n$ equations $g(x) = 0 \in \mathbb{R}^n$, singularities correspond to the points $x$ where the Jacobian matrix $J(x) = Dg(x)$ has its rank reduced from the maximal over $X$. When the rank of the Jacobian is maximal, the point is instead said to be *regular*.

In the case of the invariant set, the Jacobian matrix is computed by taking the derivative of $\tilde{B}\theta^2 - c$ w.r.t. $\theta$, resulting in

$$J_G(\theta) = 2\tilde{B}\text{diag}(\theta). \tag{4}$$

From Equation (4), we see how the Jacobian matrix has the same structure of the graph incidence matrix $\tilde{B}$, with each of the edges (the columns) weighted by its associated parameter $\theta$. Therefore, it follows that the only way in which the matrix can have a lower rank is when some of the values of $\theta$ are 0, i.e., when some of the edges are cut from the graph.

**Theorem 2** (Singularities disconnect neurons). *If $G'$ is the undirected graph obtained from $G$ by gluing together input and output nodes and neglecting edge direction, and $\mathcal{E}(\theta) \subseteq E$ is the set of edges with zero weight $\theta_e = 0 \iff e \in \mathcal{E}$, then*

$$\text{rank } J_G(\theta) = |\tilde{V}| + 1 - CC(G'_{-\mathcal{E}(\theta)}),$$

*where $CC(G)$ is the number of connected components of $G$.*

*Proof.* The proof uses tools from discrete topology and is performed by relating the rank of $J_G(\theta)$ to the kernel of a weighted graph Laplacian. The derivation can be found in Section A.5. □

Therefore, it follows that $\text{rank } J_G(\theta) = |\tilde{V}|$ for regular parameters, and its rank decreases for parameters whose zero edges disconnect some hidden neurons from both input and output. If a group of neurons is such that it is not connected by any path to both input and output neurons, it means that it is a useless component of the network as it takes no part in any computation, both in the feed-forward and in the back-propagation phases. Singularities, therefore, correspond to effective *sub-networks* of the original neural network, as shown on Figure 3a. This echoes similar results connecting sub-networks to the singular points of the neuromanifold (Trager et al., 2019; Shahverdi, 2024; Arjevani et al., 2025).

This observation can be leveraged to prove the following result.

**Proposition 5** (Singularities are invariant under GF). *Let $\theta(t_0) \in \mathcal{H}_G(c)$ and $W \subseteq \tilde{V}$ be a disconnected set of nodes at time $t_0$, that is, $\theta_{(u,v)}(t_0) = 0$ for $(u, v) \in E$ with $u \in W$ and $v \notin W$ or vice versa. Then, at a later time $t > t_0$, $\theta_{(u,v)}(t) = 0$ still holds and $W$ is still disconnected.*

*Proof.* Intuitively, the result is proved by noticing that, in a singular $\theta$, the activation of the neurons in $W$ will be 0, meaning that backpropagation will assign zero gradients to all the edges $(u, v), v \in W \wedge u \in W$. The full proof can be found in Section A.6 $\qquad\square$

Proposition 5 means that if the parameter reaches a singularity, then it cannot escape from it: once a network module has been killed, it cannot be revived.

**Singularities are rare.** One would be tempted to think that the presence and invariance of singularities could provide the explanation for the neural network performing an automatic model selection through the progressive movement from one singularity to another, smaller one. We show here, however, that this picture does not hold for the singularities of the invariant set for two reasons: **1.** Given a random initialization, the probability of $\mathcal{H}_G(c)$ having singularities is 0. **2.** If $\mathcal{H}_G(c)$ has singularities, then the learning trajectory cannot reach them in finite time.

**Proposition 6.** *Let $c \in \mathbb{R}^{|\tilde{V}|}$. If $\mathcal{H}_G(c)$ admits singularities then there exists a subset of hidden neurons $W \subseteq \tilde{V}$ such that $\sum_{v \in W} c_v = 0$.*

*Proof.* By definition, a singularity identifies a disconnected set of nodes $W \subseteq \tilde{V}$. If we denote the edges inside $W$ by $E_W = \{e = (u, v) \in E \mid u, v \in W\}$, we have that $\sum_{v \in W} c_v = \sum_{e \in E_W} \theta_e^2 - \theta_e^2 = 0$. This is because each edge in $E_W$ is shared by exactly 2 nodes of $W$ and all other edges in or out of $W$ have weight 0. $\qquad\square$

To have a singularity, therefore, an exact equality condition on $c$ must hold (Figure 3). If we sample the initial parameter with any initialization scheme where each parameter is independently sampled from the real numbers $\mathbb{R}$, we see that the probability that a set of neurons will have sum *exactly* zero must be zero. Moreover, a stronger statement than the one of Proposition 5 can be derived.

**Proposition 7.** *Under GF optimization, we have that* $\operatorname{rank} J(\theta(0)) = \operatorname{rank} J(\theta(t)) \ \forall \ 0 \leq t < \infty$.

This result (Section A.7) tells us that a gradient flow trajectory cannot fall into a singularity in finite time. Together, Propositions 6 and 7 implies that singularities generally don't exist in the optimization space when using common initializations and, even with a specifically chosen initialization ($c = 0$), they are effectively unreachable under gradient flow.

**Inducing singularities.** As shown above, singularities can allow the model to perform "self-pruning" but they are in general hard to reach. To actively drive the training dynamics towards them, we explore the use of regularization. An underlying motivation is the application of differentiable pruning in a very general way, entirely agnostic to the DAG topology. The idea is illustrated by Figure 3b.

A natural approach to target the singularities given by Theorem 2 is to directly penalize the number of neurons which are connected through paths to the input or output. To formalize this, we can leverage the Jacobian of $\mathcal{H}_G(c)$, $J_G(\theta) = 2\tilde{B}\operatorname{diag}(\theta)$. Promoting singularities corresponds to maximizing the dimension of the tangent space of the invariant set, which translates to minimizing the rank of $J_G(\theta)$. Since the rank is a non-differentiable function, we instead penalize the *nuclear norm* $\|J_G(\theta)\|_*$—the sum of its singular values—as a smooth surrogate (Zhao, 2012).

**Numerical experiments** As an illustrative example, we test our approach on the Breast Cancer dataset (Wolberg et al., 1993) using a range of MLP architectures—shallow and deep, with or without biases and skip connections. To approximate continuous gradient flow, we train with SGD using a small step size (0.001), comparing nuclear norm regularization against L1 and L2. Tracking the Jacobian rank during training confirms that the nuclear norm consistently drives the model toward singularities (fig. 3c). For instance, a shallow network can disconnect around 18 of its 20 hidden units, whereas L2 and unregularized training leave all neurons active. Surprisingly, L1 regularization performs similarly to the nuclear norm, despite being traditionally associated with parameter (not neuron) sparsity. This suggests that L1 may implicitly promote singularities, and, while a precise theoretical understanding is outside the scope of this paper, we provide an empirical analysis in Section A.10. Finally, reaching singular configurations guarantees the ability to perform lossless pruning, and while L2 is already quite robust to pruning, disconnecting active neurons introduces modifications to the implemented function (fig. 3d). All experimental details can be found in Section A.8, additional experiments in Section A.9.2.

## 4 CONCLUSION

In this work, we investigated two underexplored pathologies in the training of ReLU neural networks defined over directed acyclic graphs (DAGs): the (dis)connectedness of the invariant parameter space and the emergence of singularities within it. By leveraging the symmetry properties of homogeneous activations and analyzing the associated conservation laws under gradient flow, we provided a complete characterization of the invariant set as an algebraic variety constrained by quadratic balance conditions.

Our topological analysis revealed that disconnections in the optimization space are dictated by the presence of bottleneck neurons and an imbalance in flow conditions. We further demonstrated that singularities correspond to effective subnetworks, and although gradient flow trajectories cannot reach them in finite time, their role can be leveraged to improve structured pruning. We introduced a nuclear norm regularizer that promotes convergence toward such configurations. Surprisingly, we observed that L1 regularization can induce comparable effects, hinting at a deeper connection between sparsity and singularity-driven pruning.

**Limitations.** The limitations of our work mainly lie in the fact that the theoretical analysis is fully based on the assumption of using neural networks with homogeneous activation functions trained with gradient flow on an unregularized loss. Other training algorithms (like Adam) and regularizers are not subject to the same conservation laws described here. Likewise, in discrete settings the conservation laws only hold approximately with bigger stepsizes, incurring in bigger violations of the laws.

ACKNOWLEDGMENTS

M.N. acknowledges the project PNRR-NGEU, which has received funding from the MUR – DM 352/2022. G.P. acknowledges partial support by ERC Consol idator Grant RUNES (Grant no. 101171380) and the MSCA Doctoral Network BeyondTheEdge(Grant no. 101120085). This study was carried out within the FAIR - Future Artificial Intelligence Research and received funding from the European Union Next-GenerationEU (PIANO NAZIONALE DI RIPRESA E RESILIENZA (PNRR) – MISSIONE 4 COMPONENTE 2, INVESTIMENTO 1.3 – D.D. 1555 11/10/2022, PE00000013). This manuscript reflects only the authors' views and opinions; neither the European Union nor the European Commission can be considered responsible for them.

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

# A APPENDIX / SUPPLEMENTAL MATERIAL

## A.1 LLM USAGE

LLMs (ChatGPT) were used to aid in polishing the paper after writing.

## A.2 GEOMETRIC DERIVATION OF THE NETWORK FLOW EQUATION

Let $\theta \in \mathbb{R}^{|E|}$ be a parameter configuration. Let us focus on a single hidden neuron $v$, to which we associate the neuron-wise rescaling action $T_\alpha^v(\theta)$ that, for any $\alpha > 0$ acts as $T_\alpha^v(\theta)_{(i,v)} = \alpha\theta_{(i,v)}$, $T_\alpha^v(\theta)_{(v,j)} = \frac{1}{\alpha}\theta_{(v,j)}$ on the parameters associated to the edges coming in and out of $v$, respectively, and leaves the other elements of $\theta$ unchanged. $T^v\theta = \{T_\alpha^v(\theta) : \alpha \in \mathbb{R}_+\}$ is the orbit of $\theta$ under rescaling of the neuron $v$. It follows from the fact that we are dealing with ReLU networks that the loss function $L$ is constant over $T^v\theta$, $L(T_\alpha^v\theta) = L(\theta) \; \forall \alpha > 0$.

If $\theta$ is not zero on all the edges coming in and out of $v$, we have that the action $T^v$ is *free*, meaning that $T_\alpha^v\theta = \theta \iff \alpha = 1$, implying that the orbit $T^v\theta$ is diffeomorphic to $\mathbb{R}_+$. Therefore, the orbit $T^v\theta$ is a smooth manifold admitting a tangent space at $\theta$.

The fact that $L$ is constant over $T^v\theta$ means that $T^v\theta$ is contained into the level set of $L$ at $\theta$. The gradient of a function at a point is always orthogonal to the level set it is contained in, meaning that, in particular, the gradient $g(\theta)$ will be orthogonal to the tangent space of $T^v\theta$ at $\theta$.

To derive a vector generating this 1-dimensional tangent space, it is enough to consider the equation describing $T^v\theta$, differentiate w.r.t. $\alpha$ and evaluate at $\alpha = 1$.

$$\left(\frac{d}{d\alpha}\bigg|_{\alpha=1} T_\alpha^v(\theta)\right)_{(i,v)} = \frac{d}{d\alpha}\bigg|_{\alpha=1} \alpha\theta_{(i,v)} = \theta_{(i,v)} \tag{1}$$

$$\left(\frac{d}{d\alpha}\bigg|_{\alpha=1} T_\alpha^v(\theta)\right)_{(v,j)} = \frac{d}{d\alpha}\bigg|_{\alpha=1} \frac{1}{\alpha}\theta_{(i,v)} = -\theta_{(i,v)} \tag{2}$$

$$\left(\frac{d}{d\alpha}\bigg|_{\alpha=1} T_\alpha^v(\theta)\right)_{(i,j)} = 0 \text{ if } v \notin \{i,j\}. \tag{3}$$

$$\tag{4}$$

Orthogonality of the gradient w.r.t. this vector can be written as

$$\left(\frac{d}{d\alpha}\bigg|_{\alpha=1} T_\alpha^v(\theta)\right)^\top g(\theta) = 0 \tag{5}$$

$$\sum_{i:i\to v}\theta_{(i,v)}g(\theta)_{(i,v)} - \sum_{j:v\to j}\theta_{(v,j)}g(\theta)_{(v,j)} + \underbrace{\sum_{(i,j)\in E:v\notin\{i,j\}} 0 \cdot g(\theta)_{(i,j)}}_{=0} = 0 \tag{6}$$

$$\langle\!\langle\theta, g(\theta)\rangle\!\rangle_v = 0. \tag{7}$$

## A.3 PROOF OF PROPOSITION 3

We start from the following result from de Medrano (2023).

**Proposition 8** ((de Medrano, 2023) Proposition 4.1). *The following are equivalent:*

1. $\mathcal{H}(c)$ *is non-empty.*

2. $c$ *lies in the convex cone generated by the columns of* $\tilde{B}$, $c \in \text{Co}(\tilde{B})$.

The non-emptiness of the invariant set is then completely described by the following proposition.

**Proposition 9.** $\text{Co}(\tilde{B}) = \mathbb{R}^{|\tilde{V}|}$

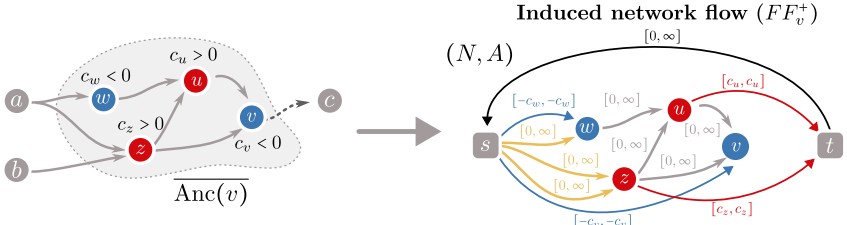

Figure 4: Visualization of the induced network flow problem $(FF_v^+)$ at a bottleneck node $v \in V_B^+$. In the right panel, we depict the internal arcs in gray, the incoming arcs in orange, the source arcs in blue, the sink arcs in red and the circulation arc in black.

*Proof.* To show that, for any $c \in \mathbb{R}^{|\tilde{V}|}$, the system $\tilde{B}x = c$ admits a non-negative solution $x$, we take a constructive approach and build $x$ explicitly.

Let us initialize $x_0 = \mathbb{1}$ by assigning the value 1 to every edge $(x_0)_e = 1 \ \forall e \in E$. Let us define the balance vector at initialization, $c_0 = \tilde{B}\mathbb{1}$ which, in general, will be different from $c$, $c_0 \neq c$. For each hidden neuron $v \in \tilde{V}$, we make an adjustment to $x_0$ that corrects the balance value at $v$.

If $(c_0)_v < c_v$, define $\delta_v = c_v - (c_0)_v$. Let us consider a path $p_v$ going from the input neurons $V_I$ to $v$ (which exists by definition of $G$) and the vector $\mathbb{1}_{p_v} \in \mathbb{R}^{|E|}$ supported on it, that is such that it assigns value 1 to each edge in $p_v$ and 0 to all other edges. Consider now $x' = x_0 + \delta_v \mathbb{1}_{p_v}$, which is non-negative because $\delta_v \geq 0$. The balance $c'_w := (\tilde{B}x')_w$ will remain unchanged in all nodes $w$ different from $v$, $c'_w = (c_0)_w$ because, in such nodes, the same quantity is added to one edge coming in and one edge going out. In $v$, the new balance value will be

$$c'_v = (\tilde{B}x')_v = (\tilde{B}x_0)_v + (\tilde{B}\mathbb{1}_{p_v})_v = (c_0)_v + \delta_v = (c_0)_v + c_v - (c_0)_v = c_v.$$

If instead $(c_0)_v > c_v$, we define $\delta_v = (c_0)_v - c_v$ and add the vector $\mathbb{1}_{p_v}$ supported on a path $p_v$ that connects $v$ to the output neurons $V_O$. Once again, in all nodes except for $v$ the balance is left unchanged, while in $v$

$$c'_v = (\tilde{B}x')_v = (\tilde{B}x_0)_v + (\tilde{B}\mathbb{1}_{p_v})_v = (c_0)_v - \delta_v = c'_v - ((c_0)_v - c_v) = c_v.$$

Repeating this process for any node $v \in \tilde{V}$ allows us to correct the balance $c'_v = c_v$ at all nodes, giving us a non-negative solution $x'$ to the system $\tilde{B}x' = c$. $\square$

## A.4 Proof of Theorem 1

The problems associated with network flow have been extensively studied in the literature (Ford and Fulkerson, 1962). The idea of the proof is to map the statement to a network flow feasibility problem and then apply well-known results in the field.

We first report here the more general version of the result from de Medrano (2023) that we reported in Proposition 4.

**Proposition 10** ((de Medrano, 2023) Proposition 4.7)**.** *The following are equivalent:*

1. $\mathcal{H}_G(c)$ *is connected.*

2. $c \in \mathrm{Co}(\tilde{B}_{-i})$ *for all $i$, where $\tilde{B}_{-i}$ denotes the matrix $\tilde{B}$ with the $i$-th column removed and $\mathrm{Co}(A)$ is the convex cone generated by the columns of $A$.*

### A.4.1 Induced network flow problem

Let us consider a general feed-forward network as in Section 2 and an out-bottleneck node $v \in V_B^+$. We will prove that connectedness is equivalent to the existence of solutions to a network flow problem associated with each bottleneck node. We now build a flow feasibility problem on a subgraph of $G$ in "circulation form", using the notation of Fathabadi and Ghiyasvand (2007).

Let $v \in V_B^+$, we construct a directed multi-graph $G_v = (N_v, A_v)$ in the following way.

The node set is $N_v = \overline{\mathrm{Anc}(v)} \sqcup \{s, t\}$, where $s$ and $t$ are two extra nodes not from $G$. The set of arcs $A_v$ contains the following sets of edges:

- the *internal arcs* $A_o = \left\{ e = (v_1, v_2) \in E \mid v_1, v_2 \in \overline{\mathrm{Anc}(v)} \right\}$;

- the *incoming arcs* $A_i = \left\{ (s, v_2) \mid \exists (v_1, v_2) \in E \text{ with } v_1 \notin \overline{\mathrm{Anc}(v)}, v_2 \in \overline{\mathrm{Anc}(v)} \right\}$;

- the *source arcs* $A_s = \left\{ (s, w) \mid w \in \overline{\mathrm{Anc}(v)}, c_w < 0 \right\}$;

- the *sink arcs* $A_t = \left\{ (w, t) \mid w \in \overline{\mathrm{Anc}(v)}, c_w > 0 \right\}$;

- the *circulation arc* $(t, s)$, to adopt the problem's "circulation form" (Fathabadi and Ghiyasvand, 2007).

Therefore $A_v = A_o \sqcup A_i \sqcup A_s \sqcup A_t \sqcup \{(t, s)\}$.

In the general network flow problem, each arc $e = (u, w)$ is assigned a lower bound $l_e = l_{uw}$ and a possibly infinite upper bound $m_e = m_{uw}$ on the possible values of the flow on them. In our case, we fix lower and upper bounds as follows. The flow on the internal and incoming arcs and the $(t, s)$ arc are required to be non-negative: $l_e = 0$ and $m_e = \infty$ for $e \in A_o \cup \{(t, s)\}$. The source arcs are constrained to carry a flow equal to the negative of the $c$ value of their endpoints: $l_e = m_e = -c_w \ \forall e = (s, w) \in A_s$. The sink arcs are constrained by the $c$ value of their starting point: $l_e = m_e = c_w \ \forall e = (w, t) \in A_t$.

The network $(N_v, A_v)$ is said *feasible* if there exists a real function on the edges, called *flow* and denoted $f : A \to \mathbb{R}$ such that

1. at each node the flow is conserved $\sum_{u:(u,w)\in A_v} f_{(u,w)} - \sum_{z:(w,z)\in A} f_{(w,z)} = 0 \ \forall w \in N$;

2. for each edge the bounds are respected $l_e \le f_e \le m_e \ \forall e \in A_v$.

Remembering that all this construction is built around an out-bottleneck $v$, we call this problem $(FF_v^+)$ for "flow feasibility at $v \in V_B^+$" and we visualize the construction in Figure 4. The analogous problem for an in-bottleneck $w \in V_B^-$ is converted to the same formulation by reversing arrows in the DAG and changing the sign of the $c$ vector, we call it $(FF_w^-)$.

### A.4.2 Translation of the characterization by network flow problems

Having defined the induced problems $(FF_v^+)$ and $(FF_w^-)$, we have a first equivalence lemma:

**Lemma 1.**

$$\mathcal{H}_G(c) \text{ connected} \iff \forall v \in V_B^+, (FF_v^+) \text{ has a solution and}, \forall w \in V_B^-, (FF_w^-) \text{ has a solution.} \tag{8}$$

*Proof.* ( $\implies$ ) We start with the forward implication, assuming that the invariant set is connected and building a solution for every flow feasibility problem associated to bottleneck neurons.

Assume that $\mathcal{H}_G(c)$ is connected. Then, by the characterization of connectedness in Proposition 10 we know that $c \in \mathrm{Co}(\tilde{B}_{-i}) \ \forall i$. This tells us that removing any column $i$ from $\tilde{B}$, there still exists an $x \in \mathbb{R}_+^{|E|-1}$, such that $\tilde{B}_{-i} x = c$.
We let $v \in V_B^+$ and ask whether $(FF_v^+)$ has a solution. Recall that there is a one-to-one correspondence between columns of $\tilde{B}$ and edges of the DAG.

We choose the column $i$ to remove to be the one corresponding to the unique out-edge $e^*$ of $v$ and get a solution vector $x^* \in \mathbb{R}_+^{|E|-1}$ to $\tilde{B}_{-i} x^* = c$.

$x^*$ can be seen as a function: $x^*: E \smallsetminus \{e^*\} \to \mathbb{R}_+$. Using $x^*$, we explicitly build a solution T the problem $(FF_v^+)$ with node and arc sets $(N, A)$ as described above.

$$f: A \to \mathbb{R}_+$$
$$e \mapsto \begin{cases} x^*(e) & \text{if } e \in A_o \sqcup A_i \\ -c_u & \text{if } e = (s, u) \in A_s \\ c_u & \text{if } e = (u, t) \in A_t \\ \sum_{e \in A_t} f_e(e) & \text{if } e = (t, s) \end{cases} \qquad (9)$$

Note that we used the fact that incoming edges in $G_v = (N_v, A_v)$ can be mapped one-to-one with edges from $V \smallsetminus \overline{\mathrm{Anc}(v)}$ to $\overline{\mathrm{Anc}(v)}$. To check that this flow is feasible, we have to check the two conditions of conservation and boundedness. For $u \in N \smallsetminus \{s, t\}$, the conservation of flow is assured by the fact that $\sqrt{x^*}$ is a member of $\mathcal{H}_G(c)$ and therefore respects the balance conditions with balance $c_u$. This value $c_u$, depending on its sign, is accounted for in $G_v$ by the arcs going to $s$ or $t$. For $t$, conservation is assured by the definition of $f$ and for $s$ it is assured by summing conservation equations for all nodes in $N \smallsetminus \{s, t\}$.

Checking the boundedness condition is immediate for $e \in A_o \sqcup A_i$, since $x^*$ has only non-negative values. For source and sink arcs, by definition of $f$ they are set with the only possible value, and $f_{(t,s)}$ is non-negative as a sum of non-negative terms.

The same reasoning can be applied for any in-bottleneck $v \in V_B^-$.

$(\Longleftarrow)$ We now prove the reverse implication and assume $\forall v \in V_B^+, (FF_v^+)$ has a solution and analogously for $(FF_v^-)$. By Proposition 3, we know that $\mathcal{H}_G(c)$ is non-empty. To show that it is connected, we need to show that $c \in \mathrm{Co}(\tilde{B}_{-i}) \; \forall i$.

Stated differently, we need to show that when removing an edge, we can still find a solution $\theta^* \in \mathbb{R}^{|E|-1}$ that satisfies every node's balance condition in the DAG.

Let us pick an edge $e^* = (v, v') \in E$, remove it, and distinguish 3 cases.

**1.** If $v \notin V_B^+$ and $v' \notin V_B^-$, the new DAG still has the property that each hidden node is contained in one path from input $V_I$ to output $V_O$, so we can apply the non-emptiness property on this new DAG and get a solution.

**2.** If $v \in V_B^+$ or $v' \in V_B^-$ but not both, we focus on $v \in V_B^+$. We will construct a function $x: E \smallsetminus \{e^*\} \to \mathbb{R}_+$, such that $\theta^* = \sqrt{x}$ respects all balance conditions in the new DAG $G_{-e^*}$.

We start by splitting the edge set in 3 parts: $E \smallsetminus \{e^*\} = E_o \sqcup E_f \sqcup E_{o \to f}$, where the edge subsets are defined as follows. For a generic edge $e = (u, w) \in E \smallsetminus \{e^*\}$, we let $e \in E_o$ if $u, w \notin \overline{\mathrm{Anc}(v)}$ i.e. $e$ is an edge whose nodes are not involved in the network flow problem $(FF_v^+)$. We let $e \in E_f$ if $u, w \in \overline{\mathrm{Anc}(v)}$, that is if both nodes are involved in $(FF_v^+)$. Lastly, $e \in E_{o \to f}$ if $u \notin \overline{\mathrm{Anc}(v)}$ and $w \in \overline{\mathrm{Anc}(v)}$ i.e. if the edge is a hybrid edge connecting a node not involved in $(FF_v^+)$ to a node involved in it. Notice that the case $u \in \overline{\mathrm{Anc}(v)}$ and $w \notin \overline{\mathrm{Anc}(v)}$ does not exist by definition of the pure ancestor set.

The next step consists in crafting 3 functions on the edges and then gluing them together to obtain a solution. By non-emptiness, again, we know that if $E_o \neq \varnothing$, we can find a solution $\theta_o^*$ on the DAG restricted to $V \smallsetminus \overline{\mathrm{Anc}(v)}$ i.e. an independent solution for the DAG where we have removed the bottleneck and its pure ancestors. We define:

$$x_o(e) = \begin{cases} 0 & \text{if } e \notin E_e \\ (\theta_o^*)^2(e) & \text{if } e \in E_e \end{cases} \qquad (10)$$

Then, by denoting with $f^*$ a solution of $(FF_v^+)$, we define:

$$x_f(e) = \begin{cases} 0 & \text{if } e \notin E_f \\ f^*(e) & \text{if } e \in E_f \end{cases} \qquad (11)$$

Finally, for the hybrid edges in $E_{o\to f}$, we leverage the fact that they can be mapped one-to-one to the *incoming* edges in $(FF_v^+)$. Just like we did for $E_f$, we thus assign to each one of them the value of the solution of the network flow problem. Let us denote $x_{o\to f}$ the function which does this assignment for edges in $E_{o\to f}$ and is zero elsewhere.

At this point, the function $\theta\colon e \mapsto \sqrt{x_o(e) + x_f(e) + x_{o\to f}(e)}$ respects balance conditions for nodes in $\overline{\mathrm{Anc}}(v)$, this is assured by the $x_f + x_{o\to f}$ part being a solution to $(FF_v^+)$. It also respects the balance for nodes which are not pure ancestors and are not connected to pure ancestors, by the definition of $x_o(e)$.

For the other nodes $u \in \tilde{V} \smallsetminus \overline{\mathrm{Anc}}(v)$ such that $\exists (u,w) \in E$ with $w \in \overline{\mathrm{Anc}}(v)$, the balance might not be immediately respected. Indeed, the values assigned to the edges in $E_{o\to f}$ will disrupt the balance given by the $x_o$ part of the function.

Luckily, each of these disruptions may be resolved locally, without influencing the balance of the other nodes. Given any $u \notin \overline{\mathrm{Anc}}(v)$, we compute its balance $c'_u = \sum_{v:(v,u)\in E\smallsetminus\{e^*\}} \theta^2_{(v,u)} - \sum_{w:(u,w)\in E\smallsetminus\{e^*\}} \theta^2_{(u,w)}$.

If $c'_u > c_u$, we pick any path $p$ in $G_{-e^*}$ from $u$ to the output nodes $p = (p_1, \ldots, p_n), p_1 = u, p_n \in V_O$. Let $\mathbb{1}_p$ be the indicator function which assigns 1 to the edges in $p$ and 0 to the other edges. Note that this path exists because $u$ is not a pure ancestor of $v$ and thus removing $e^*$ does not disconnect $u$ from the output nodes. If we define $\theta' = \sqrt{\theta^2 + (c'_u - c_u)\mathbb{1}_p}$, we see that the balance of $\theta'$ at $u$ is

$$\sum_{v:(v,u)\in E\smallsetminus\{e^*\}} \theta'^2_{(v,u)} - \sum_{w:(u,w)\in E\smallsetminus\{e^*\}} \theta'^2_{(u,w)} = c'_u - c'_u + c_u = c_u,$$

while it is left unchanged at any other node in the path because the quantity added to its inputs is the same as the one added to the outputs.

If $c'_u < c_u$ instead, we can pick any path $p$ in $G_{-e^*}$ from input nodes to $u$, $p = (p_1, \ldots, p_n), p_1 \in V_I, p_n = u$ and define $\theta' = \sqrt{\theta^2 + (c_u - c'_u)\mathbb{1}_p}$ to fix the balance at $u$.

Repeating this process for every hidden node, we are able to find a function $\theta^*$ on the edges of $G_{-e^*}$ which satisfies the balance equation at every node, meaning that $\tilde{B}_{-e^*}(\theta^*)^2 = c$.

**3.** If $v \in V_B^- \cap V_B^+$, we have that $\overline{\mathrm{Anc}}(v) \cap \overline{\mathrm{Desc}}(v) = \{v\}$ and so no edge is shared between the two sets of nodes and we can deal with $v$ being an in- and out-bottleneck independently. $\qquad\square$

Now that we have a characterization of connectedness in terms of flow feasibility we move on to study this feasibility with the positive cut method. To avoid cluttering, we refer to $(FF_v)$ to denote either $(FF_v^+)$ or $(FF_v^-)$ as these are the same problem, differing only in their origin.

We now resort to the following classic result:

**Proposition 11** (Hoffman's theorem (Hoffman, 1958), reported in Fathabadi and Ghiyasvand (2007))**.** *A network with conservation and boundedness constraints with non-negative lower bounds is feasible if and only if for every non-trivial partition $(S,T)$ $S \sqcup U = N$, called a* cut*, we have $V(S,T) \le 0$, where:*

$$V(S,T) = \sum_{\substack{i\in S \\ j\in T}} l_{ij} - \sum_{\substack{i\in T \\ j\in S}} m_{ij} \tag{12}$$

Equivalently, we have that $(FF_v)$ does not have a solution if and only if there exists a strictly positive cut:

$$(FF_v) \text{ does not have a solution} \Leftrightarrow \exists S,T \subset N, S,T \ne \varnothing, S \cap T = \varnothing, S \cup T = N, \quad V(S,T) > 0 \tag{13}$$

**Lemma 2.** *Let $(S,T)$ be a strictly positive cut i.e. $V(S,T) > 0$. Then*

$$\{s,t\} \subset S \text{ OR } \{s,t\} \subset T$$

*Proof.* If $s \in S$, then $t \in S$ otherwise the arc $(t,s)$ goes from $T$ to $S$ and it has infinite upper bound. If $s \in T$, then $t \in T$ otherwise there is no arc with strictly positive lower bound entering $T$ and $V(S,T) \le 0$. $\qquad\square$

**Lemma 3.** *Let $v$ be the bottleneck of the problem $(FF_v)$ and $G_v = (N, A)$ the induced network flow problem. Let $(u, w) \in A$ with $u, w \in N \setminus \{s, t\}$. Then*

$$V(S, T) > 0 \Rightarrow \begin{cases} u \in T \Rightarrow w \in T \\ w \in S \Rightarrow u \in S \end{cases} \tag{14}$$

*We say that $T$ is* forward closed *and $S$ is* backward closed.

*Proof.* We have that $N \setminus \{s, t\} = \overline{\mathrm{Anc}}(v)$. This means that $(u, w)$ is an internal arc so $m_{uw} = \infty$. Both statements are proved by observing that there cannot be an internal arc starting in $T$ and ending in $S$, otherwise $V(S, T) = -\infty$. In other words an arc starting in $T$ must end in $T$, and an arc ending in $S$ must start in $S$. □

**Lemma 4.** *Let $(S, T)$ be a strictly positive cut. Then*

$$\{s, t\} \subset S \tag{15}$$

*Proof.* From Lemma 2, we know $\{s, t\} \subset S$ OR $\{s, t\} \subset T$.
Let $v$ be the bottleneck of the problem $(FF_v)$ and $G_v = (N, A)$ the induced network flow problem. We proceed by contradiction and suppose that $\{s, t\} \subset T$. Let us pick an arbitrary node $w \in N \setminus \{s, t\} = \overline{\mathrm{Anc}}(v)$. In the DAG, any path going backward from $w$ will necessarily pass through an incoming edge as the path leave $\overline{\mathrm{Anc}}(v)$, otherwise $w$ would be disconnected from the input. Formally,

$$\forall w \in N \setminus \{s, t\}, \exists z \in N \setminus \{s, t\} \cap \mathrm{Anc}(w), (s, z) \in A_i$$

So $z$ must be in $T$ otherwise $(s, z)$ would go from $T$ to $S$ with $m_{sz} = \infty$. By the forward closure property of $T$ from Lemma 3, $w \in T$.

Since $w \in N \setminus \{s, t\}$ was picked arbitrarily, it follows that $N \setminus \{s, t\} \subset T$. By hypothesis $\{s, t\} \subset T$ and so $T = N$ which is impossible since the partition $(S, T)$ must be non-trivial. □

Finally, we prove the main theorem.

**Proposition 12.**

$$\mathcal{H}_G(c) \text{ is connected } \Leftrightarrow \begin{cases} \forall v \in V_B^+, \forall T \subset \overline{\mathrm{Anc}}(v) \text{ s.t. } T \text{ closed by forward edges, } \sum_{u \in T} c_u \geq 0 \\ \forall v \in V_B^-, \forall T \subset \overline{\mathrm{Desc}}(v) \text{ s.t. } T \text{ closed by backward edges, } \sum_{u \in T} c_u \leq 0 \end{cases} \tag{16}$$

*Proof.* Let $v \in V_B^+$.

$$(FF_v^+) \text{ has no solution} \Leftrightarrow \exists (S, T), V(S, T) > 0$$
$$\Leftrightarrow \exists (S, T), T \text{ forward closed}, V(S, T) > 0 \qquad \text{(Lemma 3)}$$
$$\Leftrightarrow \exists T \subset \overline{\mathrm{Anc}}(v) \text{ forward closed}, V(N \setminus T, T) > 0 \qquad \text{(Lemmas 4)}$$
$$\Leftrightarrow \exists T \subset \overline{\mathrm{Anc}}(v) \text{ forward closed}, \sum_{\substack{i \notin T \\ j \in T}} l_{ij} - \sum_{\substack{i \in T \\ j \notin T}} m_{ij} > 0 \qquad \text{(Equation 12)}$$

(17)

In the third row, we used the fact that $S = N \setminus T$. Now, notice that we can rewrite the first sum as $\sum_{\substack{i \notin T \\ j \in T}} l_{ij} = \sum_{\substack{u \in T \\ c_u < 0}} c_u$, because the only non-null lower bounds leaving $S$ are those of source arcs and these belong to nodes having $c_u < 0$. The second sum can be rewritten as $\sum_{\substack{i \in T \\ j \notin T}} m_{ij} = \sum_{\substack{u \in T \\ c_u > 0}} c_u$ as a consequence of the forward stability of $T$: the only edges leaving $T$ are sink arcs from $T$ i.e. from nodes having $c_u > 0$.
This allows us to rewrite the equivalence obtained above as

$$(FF_v^+) \text{ has no solution} \Leftrightarrow \exists T \subset \overline{\mathrm{Anc}}(v) \text{ forward closed}, \sum_{\substack{u \in T \\ c_u < 0}} -c_u - \sum_{\substack{u \in T \\ c_u > 0}} c_u > 0$$
$$\Leftrightarrow \exists T \subset \overline{\mathrm{Anc}}(v) \text{ forward closed}, \sum_{u \in T} c_u < 0$$

(18)

Therefore by negating both statements:

$$(FF_v^+) \text{ has a solution} \Leftrightarrow \forall T \subset \overline{\mathrm{Anc}}(v) \text{ forward closed, } \sum_{u \in T} c_u \geq 0 \qquad (19)$$

The complementary statement for in-bottlenecks is obtained by reversing the arrows and following similar steps, and we conclude by using Lemma 1:

$$\mathcal{H}_G(c) \text{ is connected} \Leftrightarrow \begin{cases} \forall v \in V_B^+, (FF_v^+) \text{ has a solution} \\ \forall v \in V_B^-, (FF_v^-) \text{ has a solution} \end{cases}$$
$$\Leftrightarrow \begin{cases} \forall v \in V_B^+, \forall T \subset \overline{\mathrm{Anc}}(v) \text{ forward closed, } \sum_{u \in T} c_u \geq 0 \\ \forall v \in V_B^-, \forall T \subset \overline{\mathrm{Desc}}(v) \text{ backward closed, } \sum_{u \in T} c_u \leq 0 \end{cases} \qquad (20)$$

$\square$

A.5   PROOF OF THEOREM 2

Let us compute $\operatorname{rank} J_G(\theta)$.

First, observe that if $\theta_e \neq 0 \ \forall e \in E$, then $\operatorname{rank} J_G(\theta) = \operatorname{rank}(\tilde{B})$ as $\operatorname{diag}(\theta)$ is invertible. Therefore, the rank of the Jacobian can decrease if and only if some parameters are 0, i.e. when some edges are effectively removed from the computational graph.

Define now $\operatorname{diag}(\theta)_{ee}^{\dagger} = 1/\theta_e$ if $\theta_e \neq 0$ else 0 and $\mathcal{E}(\theta) = \{e \in E : \theta_e = 0\}$ the set of zero-weight edges. Observe that

$$\operatorname{rank} J(\theta) = \operatorname{rank}(\tilde{B}\operatorname{diag}(\theta)) = \operatorname{rank}(\tilde{B}\operatorname{diag}(\theta)\operatorname{diag}(\theta)^{\dagger}) = \operatorname{rank}(\tilde{B}_{-\mathcal{E}(\theta)}),$$

where $\tilde{B}_{-\mathcal{E}}$ is $\tilde{B}$ with the columns corresponding to edges in $\mathcal{E}$ put to 0. Now, it holds that

$$\operatorname{rank}(\tilde{B}_{-\mathcal{E}(\theta)}) = \operatorname{rank}(\tilde{B}_{-\mathcal{E}(\theta)}^{\top}) = |\tilde{V}| - \dim \ker(\tilde{B}_{-\mathcal{E}(\theta)}^{\top}), \tag{21}$$

where the last equality follows from the rank nullity theorem.

To relate $\dim \ker(\tilde{B}_{-\mathcal{E}(\theta)}^{\top})$ to the topological properties of $G$, we briefly introduce some concepts stemming from relative homology (Hatcher, 2002).

Let $C_0(G)$ be the vector space of real functions on the nodes of $G$ (all neurons) $x \in C_0(G) \implies x{:}V \to \mathbb{R}$. These functions are customarily called the *0-chains* (of $G$). Let $C_1(G)$ be the vector space of real functions on the edges of $G$, $y \in C_1(G) \implies y{:}E \to \mathbb{R}$; these are called the *1-chains* (on $G$). We can see the incidence matrix $B$ (with the rows associated with all nodes included) as the matrix representation of the linear operator from 1-chains to 0-chains $B{:}C_1(G) \to C_0(G)$. If $\mathbb{1}_v$ is the indicator function on node $v \in V$ and $y = \sum_{e \in E} y_e \mathbb{1}_e$

$$By = \sum_{v \in V} \left( \sum_{e=(w,v), w \in V} y_e - \sum_{e=(v,u), u \in V} y_e \right) \mathbb{1}_v.$$

In this setting, $\ker B^{\top}$ is called the *0-th cohomology group* of $G$ and denoted by $H^0(G)$; and $\dim \ker B^{\top}$ is proven to be equal to the number $CC(G)$ of connected components of $G$.

Let us now pick the subset of input and output nodes $\partial V \subseteq V$ and define the space of *relative 0-chains* $C_0(G, \partial V) = \frac{C_0(G)}{C_0(\partial V)}$ i.e. the quotient space of the functions on the nodes modulo the space of functions on the input and output nodes. This means that we identify two *0-chains* $c$ and $c'$ iff $c(v) = c'(v)$ for all internal nodes $v \in \tilde{V} := V \smallsetminus \partial V$. An element in $C_0(G, \partial V)$ will thus be an equivalence class

$$[x] \in C_0(G, \partial V) \implies [x] = \left[ \sum_{v \in V} x_v \mathbb{1}_v \right] = \left[ \sum_{v \in \tilde{V}} x_v \mathbb{1}_v \right],$$

as, by definition of quotient vector space, we have that $[\mathbb{1}_v] = [0] \iff v \in \partial V$.

The incidence matrix $B$, therefore, induces a *relative incidence matrix* $\tilde{B}{:}C_1(G) \to \frac{C_0(G)}{C_0(\partial V)}$ as $\tilde{B}y = [By]$. From this, we see that

$$\tilde{B}y = [By] = \left[ \sum_{v \in V} \left( \sum_{e=(w,v), w \in V} y_e - \sum_{e=(v,u), u \in V} y_e \right) \mathbb{1}_v \right] \tag{22}$$

$$= \left[ \sum_{v \in V \smallsetminus \partial V = \tilde{V}} \left( \sum_{e=(w,v), w \in V} y_e - \sum_{e=(v,u), u \in V} y_e \right) \mathbb{1}_v \right], \tag{23}$$

meaning that the relative incidence matrix can be represented with the incidence matrix with the rows associated with nodes in $\partial V$ removed, i.e., the $\tilde{B}$ we used in the text.

In this setup $\ker \tilde{B}^{\top}$ is known as the *0-th relative cohomology group* $H^0(G, \partial V)$ of the pair $(G, \partial V)$ and, given this characterization, we can resort to Proposition 2.22 in Hatcher (2002), and deduce that

$$\dim H^0(G_{-\mathcal{E}(\theta)}, \partial V) = \dim H^0(G_{-\mathcal{E}(\theta)}/\partial V) - 1,$$

where $G/\partial G$ is the graph $G$ with nodes in $\partial V$ all glued together, and $G_{-\mathcal{E}(\theta)}$ is the graph $G$ with the edges in $\mathcal{E}(\theta)$ removed. We can therefore go back to Equation (21) and state that

$$\mathrm{rank} J(\theta) = |\tilde{V}| - \dim H^0(G_{-\mathcal{E}(\theta)}/\partial V) + 1 = |\tilde{V}| - CC(G_{-\mathcal{E}(\theta)}/\partial V) + 1,$$

meaning that the rank of the Jacobian is less than its maximum value of $|\tilde{V}|$ only when the number of connected components of the quotient graph $G_{-\mathcal{E}(\theta)}/\partial V$ is greater than 1. This happens if and only if removing edges in $\mathcal{E}(\theta)$ disconnects a set of nodes from both input and output.

### A.6 PROOF OF PROPOSITION 5

To prove the result, we will prove that, if $\theta_e = \theta_e(t_0) = 0$ for every edge $e = (u,v)$ with $u \in W, v \notin W$ or $u \notin W, v \in W$, the same holds for the gradients: $g_e(t_0) = 0$ for every edge $e = (u,v)$ as above.

Let us denote by $\mathcal{P}_{v,V_O}$ the set of all paths from node $v$ to a node in $V_O$ i.e. $p \in \mathcal{P}_{v,V_O} \iff p = (u_1, \ldots, u_{n_p})$ with $u_1 = v$, $u_{n_p} \in V_O$ and $(u_i, u_{i+1}) \in E$, for all $i$.

Let $a_v, z_v$ be the activation and pre-activation of neuron $v$, respectively

$$z_v = \sum_{(u,v) \in E} \theta_{(u,v)} a_u, \quad a_v = \sigma(z_v).$$

The chain rule allows us to decompose the gradient of the loss flowing through neuron $v$ in the contributions of all path from $v$ to an output neuron as follows:

$$\frac{\partial L}{\partial a_v} = \sum_{p=(u_1, \ldots, u_{n_p}) \in \mathcal{P}_{v,V_O}} \frac{\partial L}{\partial a_{u_{n_p}}} \prod_{i=2}^{n_p} \frac{\partial a_{u_i}}{\partial z_{u_i}} \frac{\partial z_{u_i}}{\partial a_{u_{i-1}}} \tag{24}$$

For any neuron in the disconnected set $v \in W$, it holds that any path to the output $p \in \mathcal{P}_{v,V_O}$ contains an edge $e_p$ such that $\theta_{e_p} = 0$. If we notice that $\frac{\partial z_{u_i}}{\partial a_{u_{i-1}}} = \theta_{(u_{i-1}, u_i)}$, we have that, for any path $p$, the product in Equation (24) will contain $e_p$ and therefore its value will be 0 $\frac{\partial L}{\partial a_v} = 0 \ \forall v \in W$.

Let us now prove that this implies that the edges that disconnect $W$ have zero gradient.

First, let $e = (u, w) \in E$ with $u \in V \smallsetminus W$ and $w \in W$.

$$g_e := \frac{\partial L}{\partial \theta_e} = \frac{\partial L}{\partial a_w} \frac{\partial a_w}{\partial \theta_e} = 0,$$

because $w \in W$.

Let $e = (w, v) \in E$ with $w \in W$ and $v \in V \smallsetminus W$.

$$g_e = \frac{\partial L}{\partial \theta_e} = \frac{\partial L}{\partial a_v} \frac{\partial a_v}{\partial \theta_e} = \frac{\partial L}{\partial a_v} \frac{\partial a_v}{\partial z_v} \frac{\partial z_v}{\partial \theta_e},$$

where $\frac{\partial z_v}{\partial \theta_e} = a_w$. Given that $W$ is also disconnected from the input nodes $V_I$, any node inside must have 0 activations. Therefore $w \in W \implies a_w = 0$ and $g_e = 0$.

### A.7 PROOF OF PROPOSITION 7

Let $\dot{\theta}(t) = -g(\theta(t))$ be the evolution of the parameter configuration under gradient flow.

Let $J_G(\theta) = 2\tilde{B}\mathrm{diag}(\theta)$ and let us derive the evolution equation for $J_G$.

$$\frac{\mathrm{d}}{\mathrm{d}t} J_G(t) = \frac{\mathrm{d}}{\mathrm{d}t} 2\tilde{B}\mathrm{diag}(\theta(t)) = 2\tilde{B}\mathrm{diag}(\dot{\theta}(t)) = -2\tilde{B}\mathrm{diag}(g(\theta(t))). \tag{25}$$

It turns out that we can simplify this equation by leveraging the connections between rescaling symmetries and GF dynamics. In fact, Equation 8 of Kunin et al. (2020) states that rescaling symmetries induce a relation between the gradient and the Hessian $H$ of the loss function. If $b_v$ is the transpose of the row of $\tilde{B}$ associated with $v \in \tilde{V}$, we have that

$$H(\theta)(\theta \odot b_v) + g \odot b_v = 0 \ \forall v \in \tilde{V}. \tag{26}$$

This follows because $b_v$ has a value of -1 on the edges incoming in $v$ and +1 on the edges outgoing from $v$. Gathering the identities of Equation (26) associated to all hidden neurons, we get the following matrix equation:

$$\tilde{B}\text{diag}(\theta)H(\theta) + \tilde{B}\text{diag}(g(\theta)) = 0. \tag{27}$$

Plugging Equation (27) in Equation (25), we finally get

$$\frac{\mathrm{d}}{\mathrm{d}t}J_G(t) = 2\tilde{B}\text{diag}(\theta)H(\theta) = J_G(\theta)H(\theta), \tag{28}$$

i.e. the Jacobian evolution is dictated by the Hessian matrix.

It is known (e.g. Blanes et al. (2009) page 15) that the solution of Equation (28) can be written in the following way

$$J_G(t) = J_G(0)\mathcal{T}\exp\left(\int_0^t H(s)\mathrm{d}s\right),$$

where $\mathcal{T}\exp\left(\int_0^t H(s)\mathrm{d}s\right)$ is the time-ordered matrix exponential

$$\mathcal{T}\exp\left(\int_0^t H(s)\mathrm{d}s\right) = \sum_{n=0}^{\infty}\int_0^t \mathrm{d}t_1'\int_0^{t_1'}\mathrm{d}t_2'\cdots\int_0^{t_{n-1}'}\mathrm{d}t_n'H(t_1')\cdots H(t_n')$$

which ensures that the terms of the exponential series are in the right order, as the matrices $H(t_1)$ $H(t_2)$ may not commute.

Finally, in Theorem 4 of Blanes et al. (2009), it is shown that the time-ordered matrix exponential can be written as a standard matrix exponential and is therefore invertible for any finite $t$. This means that $\text{rank}(J_G(t)) = \text{rank}(J_G(0))$, thus concluding the proof.

## A.8 Experimental details

Here we provide additional details regarding experiments described in the main paper.

### A.8.1 Disconnected neurons and pruning

Figures 3c–d show, respectively, the proportion of null singular values during training and the evolution of test loss under neuron pruning for a shallow, bias-free architecture. We report on Figure 5 results for a broader range of architectures discussed in the main text. Both nuclear norm and $L1$ regularization promoted neuron sparsity (fig. 5, left column). Note that skip connections made neuron pruning more challenging as illustrated in previous literature (Fang et al., 2023). Below, we describe our two pruning strategies; however in both cases, nuclear norm and $L1$ remain the most robust, vanilla the most fragile, and $L2$ intermediate (fig. 5, center and right columns).

**Architectural details.** We evaluate six architectures: a shallow network with 20 hidden units, with and without bias; and four multilayer perceptrons (MLPs) with three layers of 10 hidden units—varying by the presence of bias and whether a skip connection links the first and second hidden layers.

**Details on methodology.** All architectures were trained from scratch for 5000 epochs using SGD with a learning rate of $10^{-3}$. Regularization strengths were empirically tuned to balance low task loss and singular value minimization:$\alpha_{nuc} = 0.05$ (nuclear norm), $\alpha_{L1} = 10$, and $\alpha_{L2} = 20$. The total loss is defined as $L = L_{task} + \alpha_{reg} L_{reg}$. For each architecture, we sampled 30 runs that converged to low train loss models to analyze training dynamics and pruning robustness. Singular values below $10^{-3}$ were considered null. The dataset was not preprocessed. All experiments ran on CPU over approximately 50 hours.

**Pruning.** We iteratively remove entire neurons (i.e., groups of parameters) from trained models and measure the degradation in performance, as measured by the test loss. For each hidden neuron $k$, we compute a principled pruning score

$$s_k = \Big( \sum_{(i,k) \in E} \theta_{ik}^2 \Big) \Big( \sum_{(k,j) \in E} \theta_{kj}^2 \Big) \tag{29}$$

which is the product of the $L2$ norms of its input and output weights. This score is low for (nearly) disconnected neurons and invariant under neuron rescaling, making it robust to reparameterization. However, a max-based score,

$$s_k' = \text{Max}\Big( \sum_{(i,k) \in E} \theta_{ik}^2, \sum_{(k,j) \in E} \theta_{kj}^2 \Big) \tag{30}$$

produced slightly different results and is reported as well for comparison.

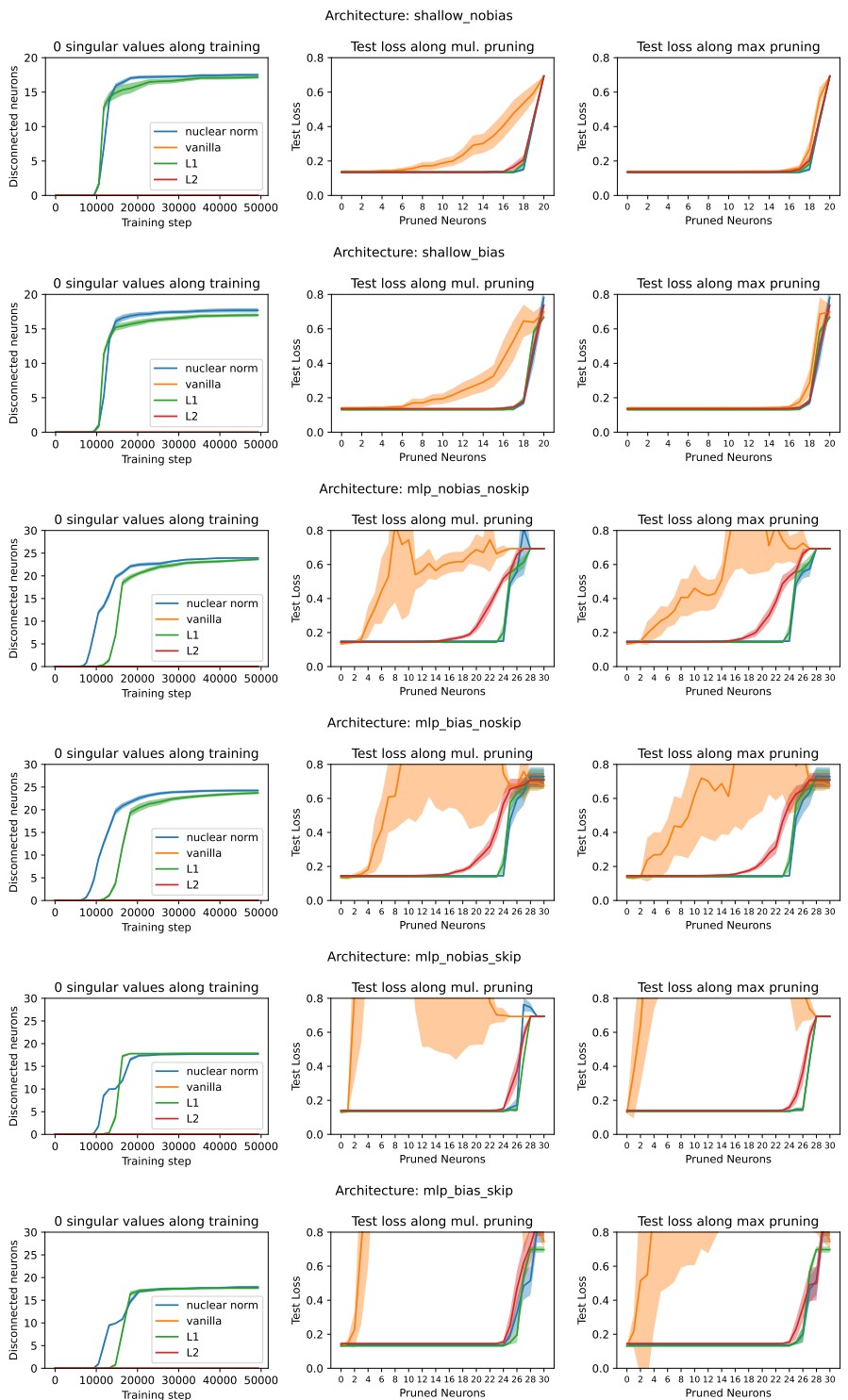

Figure 5: (**Rows**) architectures (**Left column**) number of almost 0 singular values (threshold: $10^{-3}$) along training. (**Center column**) pruning neurons on trained networks using $s_k$ (multiplicative) and (**Right column**) $s'_k$ (maximum) scores

### A.8.2 TRAINING DYNAMICS

For one training of shallow bias-free network, we present the evolution of key quantities during training to highlight how different regularizations affect their behavior.

As shown in Figure 6, both nuclear norm and $L_1$ regularization drive many neurons' balance numbers $c_k$ to zero—consistent with the fact that disconnecting a neuron in a fully connected layer requires its input and output weights to vanish, making $c_k$ null by definition (Equation (3)). Without regularization, $c_k$ values, which should be fixed under continuous gradient flow, can increase due to the effect of discrete optimization steps. These steps cause a gradual drift out of $\mathcal{H}_G(c)$, as the update vector deviates from the ideal optimization path. Notably, in this unregularized case, $c_k$ values remain more stable. Figure 7 present the complementary view of singular values.

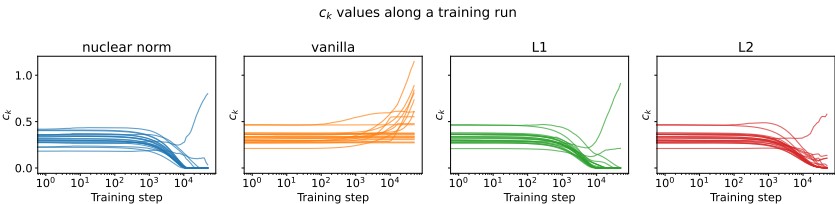

Figure 6: $c_k$ values during training of a shallow, bias-free network. $k \in [0, 20]$ is the neuron index.

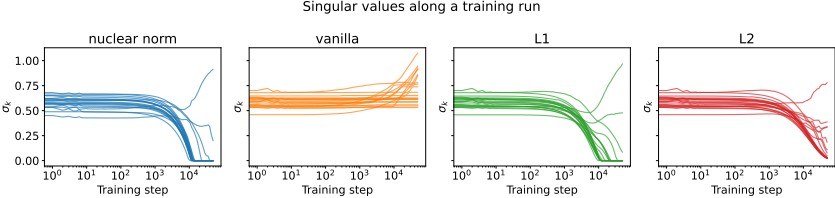

Figure 7: Singular values of the Jacobian $\tilde{B} \operatorname{diag}(\theta)$ during training of a shallow, bias-free network. $k \in [0, 20]$ is the neuron index.

## A.9    ADDITIONAL EXPERIMENTS

In this part, we present additional experiments that replicate our findings in different contexts for completeness.

### A.9.1    CONNECTIVITY

Besides the toy model presented in the paper which studies a DAG structure, we replicated connectedness results for MLPs on synthetic and real data.

**Synthetic data.**    The synthetic setup consisted of a MLP trying to learn to sum the components of a 2-dimensional input vector where each component is drawn randomly from $[0, 1]$. To solve the task, at least one of the neurons in the last layer must have a positive output weight. When the parameter space is disconnected, as stated by Corollary 2, initializing on a wrong connected component creates a pathology in which the network may be blocked from reaching the optimum, as shown on Figure 2, preventing the loss to decrease below a threshold even on the training set. For the synthetic summing task, this happens because a negative $c_k$ prevents any sign flip for a neuron. Training losses for a 10 layers, 0.5 million parameters MLP are shown on Figure 8.

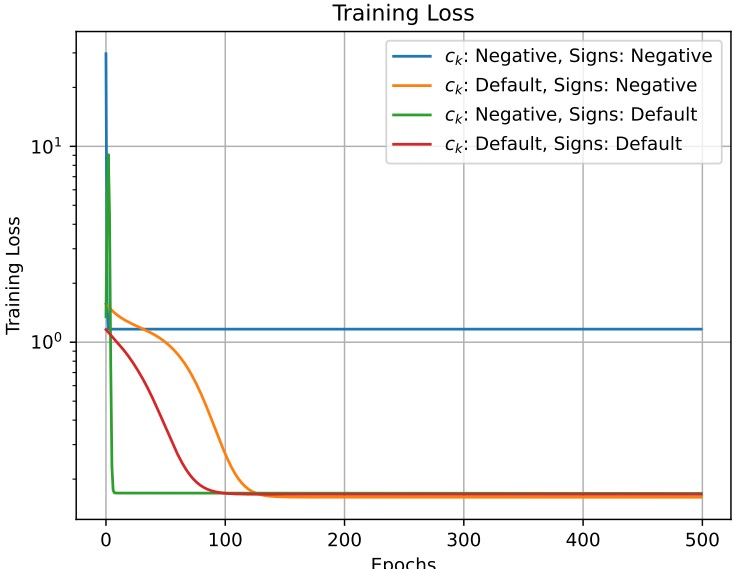

Figure 8: Training loss for 4 possible initializations of a MLP withb 10 layers. Initializations only differ in the last layer output weights. $c_k$ and signs refer to the last layer. **Blue curve**: when the training space is disconnected (negative $c_k$) and initialization is on a bad component (negative output sign for last layer neurons), the model is unable to learn correctly.

**Real data.**    We also replicate the learning obstruction on real data, both for MNIST and for the ViT features obtained from a pretrained vision transformer model (without any finetuning) on a classical high resolution dataset  (Elson et al., 2007) of cats and dogs images treated as 224 by 224 pixels and projected to 20 dimensions with UMAP. Both these tasks are set up as binary classification: the cats and dogs is already a binary dataset, while for MNIST we modify the labels to predict whether or not the digit is equal or above 5. For both datasets, it is easy for a standard MLP to achieve a low train loss as reported on Figure 9, except when the optimization space is disconnected ($c_k < 0$) and the initialization is done on a component which does not contain parameters able to predict both classes (e.g. negative output weights on last layer).

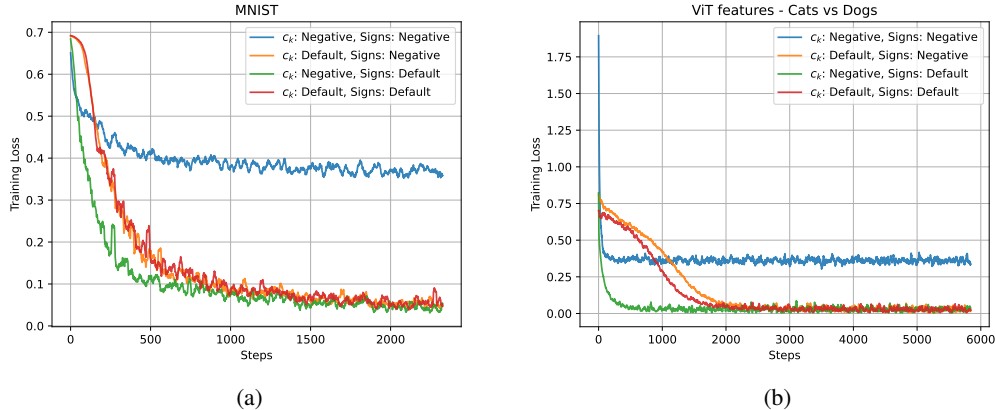

(a)                                                    (b)

Figure 9: Training losses for 4 types of initializations differing only in the last layer output weights. **(a)** Training loss on MNIST. The model is an MLP with 3 hidden layers containing 100, 50 and 10 neurons. **(b)** Training loss on ViT features extracted from a dataset of cats and dogs images. The model is an MLP with 3 hidden layers containing 20, 50 and 20 neurons.

### A.9.2 SINGULARITIES

In addition to the experiment conducted on the Brest Cancer dataset in the main paper and further described Section A.8, we obtained similar results in two others contexts: the classification of cats and dogs from ViT features discussed in Section A.9.1 and a more challenging facial attributes prediction task from face recognition model features, for which we used the CelebA dataset (Liu et al., 2015).

The ViT feature task is easily solved with almost perfect test accuracy by a three layer MLP having 2500 parameters and 90 total neurons. Adding in L1 regularization achieved 80% pruned neurons, while adding in the Jacobian regularizer around 90%, both without a significant loss of accuracy compared to the vanilla training as shown on Figure 10.

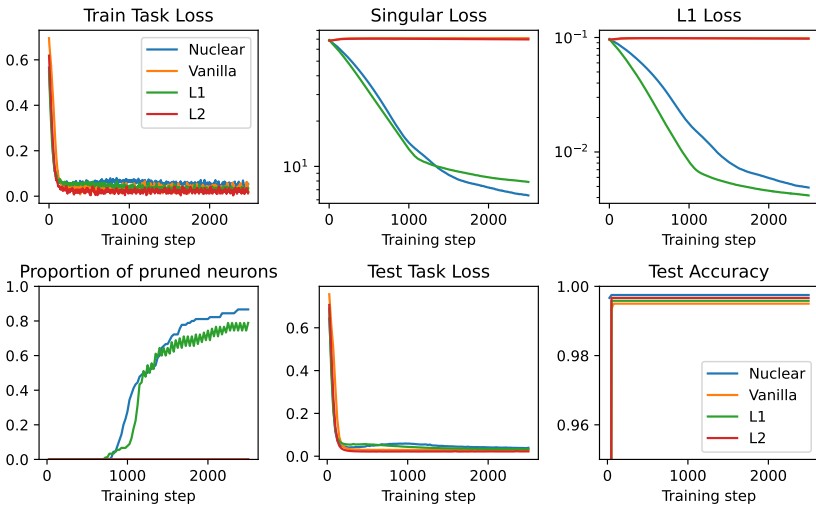

Figure 10: Key metrics along training for a binary classification task taking as input ViT features. Colored curved denote different regularizers.

For the more difficult task of predicting facial attributes like "lipstick" or "gender" from features of pretrained face recognition models, test accuracy varied depending on the attribute. However both L1 and the Jacobian regularization performed on par with the vanilla i.e. unregularized model. Depending on the attribute, L1 achieved [80-90]% neuron sparsity, while the Jacobian regularizer

achieved [85-95]% neuron sparsity. Again, no significative impact on test accuracy was observed, as reported on Figure 11.

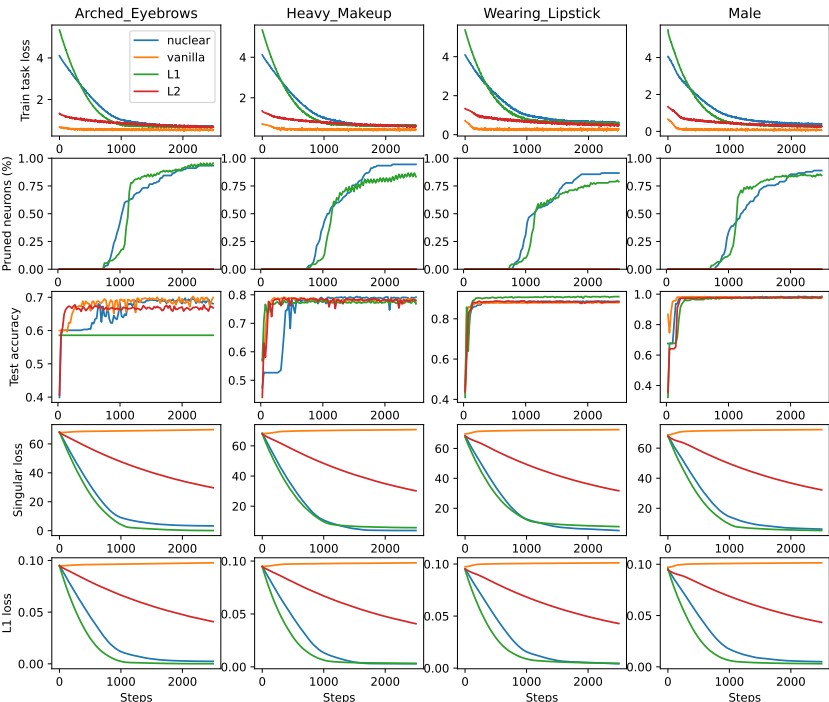

Figure 11: Key metrics along training for a prediction task aiming at predicting whether or not a facial attribute was present on a picture from features extracted with a face recognition model (here: FaceNet). Colored curves correspond to different regularizers. Dynamic pruning on row 2.

## A.10    EMPIRICAL COMPARISON BETWEEN L1 AND NUCLEAR REGULARIZATION

In the main text we observe that both the nuclear norm and L1 regularizers prune a similar number of neurons (see also Figure 3c, Figure 5, Figure 10 and Figure 11). Here, we give empirical evidence on key distinctions between these regularizers. We first show that both regularizers cannot be explained by a null model and then illustrate that the nuclear norm preserves edges on active neurons, achieving strong group regularization while L1 is more aggressive, also pruning edges belonging to active neurons.

### A.10.1    NULL MODEL

In this part we use a simple null model to rule out the explanation that L1 regularization achieves neuron sparsity solely due to its known edge sparsity mechanism. The null model works as follow: we estimate the probability $p_{L1}$ for a generic edge to be dropped after training. Intuitively, we choose an edge before training and observe after training under L1 regularization if it was dropped or not. To decide if the edge was dropped or not, we use a threshold of $10^{-3}$ which corresponds to a clear peak in the parameters values distribution, stable over multiple orders of magnitude. Then, starting from the initial computational graph (i.e. before training, with all edges), we can compute analytically the expected number of disconnected neurons if every edge is dropped with probability $p_{L1}$. Results are reported on Figure 12 left. We repeat the same analysis for the nuclear norm regularization, using the same threshold we obtain another null model with probability of dropping a random edge of $p_{nuc}$ and report the expected number of disconnected neurons on Figure 12 right. In summary, both null models cannot explain the number of pruned neurons by the number of pruned edges, meaning that in both cases there must be other underlying mechanisms. The underlying mechanism is explicit in the case of the nuclear norm regularization, since neurons are directly targeted, but remains veiled in the case of L1. Note also that the null model is an even worse explanation in the singular regularization case, indicating a stronger alternative mechanism.

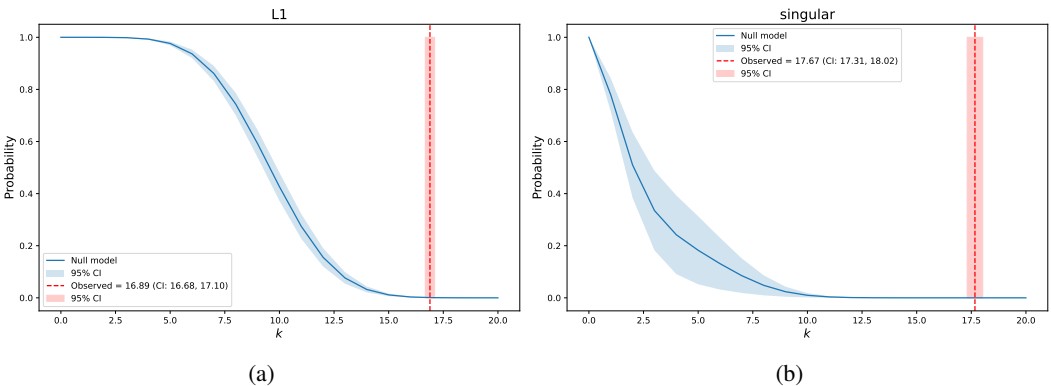

(a)                                                                                    (b)

Figure 12: Probability of having at least $k$ pruned neurons under the null model for **(a)** nuclear norm regularizer **(b)** L1 regularizer. The red dotted vertical line is the observed number of pruned neurons, the blue curve is the analytic probability of pruning at least $k$ neurons under the null model.

### A.10.2    ACTIVE WEIGHTS AND PRUNED NEURONS

To further investigate the difference between L1 and nuclear norm, we turn our attention to the distribution of parameters magnitude, which is plotted on Figure 13 left for $4$ representative trainings. For the singular regularization we observe a clear separation in the magnitude of parameters belonging to pruned and active neurons, and this is not the case for L1 regularization. This means that the nuclear norm either drops completely a neuron (i.e. all its edges at the same time) or keeps the neuron active. In contrast, the separation is less clear for L1: there are many inactive weights on active neurons. Figure 13 right shows the aggregated results for 30 models of each type, where we observe an overlap of the distribution for L1 only.

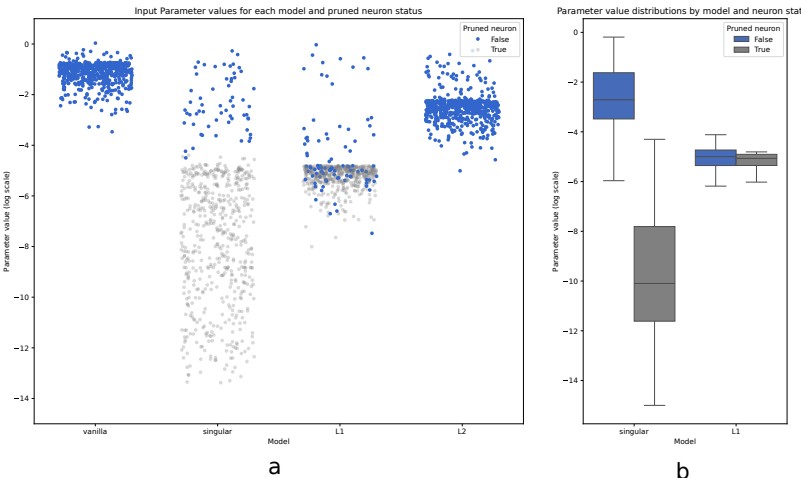

Figure 13: **a.** Absolute value of input parameters for the hidden layer of a shallow network with bias. **b.** Absolute value of parameters (including bias) for the same network architecture on 30 independent runs.

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
