# OpenReview forum: "Topology and geometry of the learning space of ReLU networks: connectivity and singularities"
_ICLR.cc/2026/Conference — ICLR 2026 Poster_

### Official Review · Reviewer_aTae · 2025-10-29

[review text omitted: it was posted to a different submission]

---

> ### Author Response · Authors · 2025-11-13
> **Issue with the review**
>
> Dear reviewer and AC,
> We'd like to respectfully point out that this review appears to be related to another paper and not ours.
> Best regards,
> Authors

---

> ### Comment · Reviewer_aTae · 2025-11-13
> **Review update**
>
> Sorry for the mistake. I updated my review correspondingly.

---

> ### Author Response · Authors · 2025-11-20
>
> Thank you for your review and for the positive assessment of our work.
> ## Weaknesses
>
> - *"I have a hard time distinguishing what are the main contributions and what are already proved in the literature. Authors might want to re-organize the section 2, and credit properly all the results (theorems, propositions, definitions) if they are ever taken/inspired by previous works."*
>
> Thank you for the feedback.
> Section $2$ blends an introduction to earlier results, their reformulation in our notation and identifies the key objects we will need in the rest of the paper. We improved the distinction between what is from literature and our work in section $2$ in the following ways:
> - We add a pointer in the dag neural network paragraph $2$ to the related work on DAG neural networks to make it clear that they have been studied.
> - We add a sentence to clarify the inclusion of biases with a reference (suggested by another reviewer).
> - We make minor modifications to improve on formulations (activation function, proposition $2$, dimensions of $f_G$) for general ease of reading.
> - We add a reference in the dag neural network paragraph to "Graph theory with applications" by Bondy about the boundary matrix, since it is a central object.
> - We add a couple of references to earlier works studying symmetries of ReLU networks.
> - We change the line before proposition $1$ to make it clearer that this is a new perspective we introduce.
> - Definition $1$ is a generalization of a definition from Nurisso et al; we inserted a citation.
> -  We add a couple lines at the beginning of section $2$ to guide the reader.
>
> We hope these edits will make the connections of our paper with previous works clearer.
> Finally, we also refer to the end of page $1$ where we list our contributions.
>
> - *"Do the author forget to define the notion of stable by forward/backward edges in the announcement of Theorem 1? Otherwise, I believe that Theorem 1 needs rephrasing to be easier to understand."*
>
> The notion was not stated explicitly, thank you for the feedback. Since stability by forward/backward edges is only used for bottleneck neurons, we propose to extend definition $2$, thereby avoiding a standalone definition.
>
> ## Questions
>
> - *"Do Proposition 6 - 7 imply that singularities are truly rare? It seems to me that the limit of GF can still be a singularity (or a sparse subnetwork). If the GF dynamics does not bias towards sparse subnetworks, do you have any idea which points are preferable for the convergence of GF?"*
>
> Proposition 6 tells us that given a random initialization, the probability that the invariant set on which learning unfolds admits singularities is zero.
> Proposition 7 adds that, even if singularities are present, for example with a carefully chosen initialization, gradient flow cannot reach them in finite time.
> Taken together, these results indicate that singularities should not be expected during standard training.
> As you point out, GF dynamics is not expected to converge to a singular point. In our paper we proposed to empirically introduce such a bias with an auxiliary loss. More generally and beyond the topic of this paper, properties of minima have been studied, for example flatness of minima has been related with generalization [1,4] (caveat: [2]) or discrete optimization [3].
>
> - [1] On Large-Batch Training for Deep Learning: Generalization Gap and Sharp Minima, Keskar et al
>
> - [2] Sharp Minima Can Generalize For Deep Nets, Dinh et al
>
> - [3] Gradient Descent on Neural Networks Typically Occurs at the Edge of Stability, Cohen et al
>
> - [4] Sharpness-Aware Minimization for Efficiently Improving Generalization, Foret et al

---

> > ### Comment · Reviewer_aTae · 2025-11-21
> >
> > Thanks the authors for the response. I will keep my current score

---

### Official Review · Reviewer_Fprd · 2025-10-30

**Soundness:** 3
**Presentation:** 3
**Contribution:** 2
**Rating:** 6
**Confidence:** 4

**Summary:**

This paper studies the feed-forward ReLU networks defined over directed acyclic graphs, examining the (dis)connectedness of the parameter space and the existence of singularities within it. The conservation laws under gradient flow are identified. Due to the disconnectedness of certain parameter configurations, certain singularities are unreachable, reducing the expressivity of ReLU networks at initialization.

**Strengths:**

- The paper is relatively well-written and polished. Illustrative figures are provided to accompany the theoretical results and aid understanding.
- The theoretical formulation is clean.
- The result on the disconnectedness of the parameter space is somewhat surprising. The implication of losing expressivity at initialization seems interesting.
- Some numerical experiments are conducted to validate theoretical results.

**Weaknesses:**

- I am wondering whether the disconnected case occurs in fully-connected ReLU networks or not, since the example network given in Figure 2(d.1) does not look like a fully-connected network. If the disconnection only occurs in networks that are not fully connected, then the statement in line 358 may be inaccurate: "the expressivity can be reduced to the extent that they lose their universal approximation capability"; because ReLU networks that are not fully connected are not universal approximators to begin with. Please feel free to correct me if I have misunderstood your results.
- In line 423, the authors state that: "given a random initialization, the probability of $\mathcal H_G(c)$ having singularities is zero." I trust that this statement itself is correct. However, it doesn't necessarily mean that the gradient flow/descent cannot go near singularities. It's quite common that ReLU networks can have saddle-to-saddle dynamics, in which the gradient flow path passes near a sequence of fixed points [1]. In those cases, even though the dynamics from random initialization never puts the parameters exactly in an invariant set, going near those fixed point is still a very prominent, if not the most prominent, trait of the learning dynamics. If I didn't misunderstand the result, the paragraph "singularities are rare" should probably come with more nuance or caveat -- "probability of having singularities being zero" doesn't mean that learning dynamics doesn't go near singularities.
- The conservation laws arising from symmetries are also studied in [2]. I am wondering how their results relate to yours results in "local conservation laws under gradient flow" in line 160.
- It might be useful to also discuss the limitation of studying gradient flow in place of SGD. Because the quantities that obey conservation laws under gradient flow can actually be time-varying in SGD [3,4].

[1] Boursier et al. "Gradient flow dynamics of shallow relu networks for square loss and orthogonal inputs." NeurIPS 2022.

[2] Ziyin. "Symmetry induces structure and constraint of learning." ICML 2024.

[3] Liu et al. "Noise and fluctuation of finite learning rate stochastic gradient descent." ICML 2021.

[4] Chen et al. "Stochastic collapse: How gradient noise attracts sgd dynamics towards simpler subnetworks." NeurIPS 2023.

**Questions:**

Is there a particular reason to use the uncommon notation of double angle brackets $《》$? I struggled to understand it from a short inline definition given in line 177. I also didn't know if this notation is essential for reading and understanding the main results.

---

> ### Author Response · Authors · 2025-11-20
>
> Thank you for the positive assessment of the work and for the insightful review.
> ## Weaknesses
> - *"I am wondering whether the disconnected case occurs in fully-connected ReLU networks or not, since the example network given in Figure 2(d.1) does not look like a fully-connected network. If the disconnection only occurs in networks that are not fully connected, then the statement in line 358 may be inaccurate: "the expressivity can be reduced to the extent that they lose their universal approximation capability"; because ReLU networks that are not fully connected are not universal approximators to begin with. Please feel free to correct me if I have misunderstood your results."*
>
> Yes, disconnections also occurs in fully-connected ReLU networks, but they are less frequent.
> Corollary $1$ and $2$ make this point precise but concretely, this can typically happen in scalar regression or binary classification settings, cases where the single output creates out-bottlenecks in the last layer. The goal of Figure 2(d.1) is to illustrate the theory using a small example where disconnection at the bottleneck $4$ depends on a remote parameter belonging to an ancestor node.
> As you noted, this example is not fully connected but in Appendix $9$ we show additional experiments using fully connected MLPs in both regression and binary-classification settings.
>
> - *"In line 423, the authors state that: "given a random initialization, the probability of having singularities is zero." I trust that this statement itself is correct.  However, it doesn't necessarily mean that the gradient flow/descent cannot go near singularities.  It's quite common that ReLU networks can have saddle-to-saddle dynamics, in which the gradient flow path passes near a sequence of fixed points [1]. In those cases, even though the dynamics from random initialization never puts the parameters exactly in an invariant set, going near those fixed point is still a very prominent, if not the most prominent, trait of the learning dynamics.  If I didn't misunderstand the result, the paragraph "singularities are rare" should probably come with more nuance or caveat -- "probability of having singularities being zero" doesn't mean that learning dynamics doesn't go near singularities."*
>
> Thank you for the interesting question.
> We would like to make sure the terminology is clear first. In saddle-to-saddle dynamics, saddles are fixed points and invariant in the sense that the gradient at a saddle is zero. This is different and complementary with our *invariant set*, which is the set of all possible trajectories and which contains saddles, minima, and on which saddle-to-saddle dynamics and learning dynamics in general happens. Likewise, singularities are not the saddles, they are subsets of the invariant set.
>
> With this in mind, we agree that gradient flow can indeed go near singularities: indeed proposition $7$ only states that gradient flow cannot *reach* a singularity in finite time.
> Studying the effect that a singularity has on a learning trajectory that passes close to it is a very interesting idea which we will definitely explore in further work.
>
> Regarding this, we point the reviewer to *Shun-ichi Amari, Hyeyoung Park, and Tomoko Ozeki. Geometrical singularities in the neuromanifold*, where the authors study some aspects of the effect that singularities in the neuromanifold of a multilayer perceptron (but not in the invariant set) have on the learning trajectories.
>
> - *"The conservation laws arising from symmetries are also studied in [2]. I am wondering how their results relate to yours results in "local conservation laws under gradient flow" in line 160."*
>
> We offer a few points of comparison between [2] and our work.
> First, in [2] they study so-called *mirror symmetries*, and they focus primarily on the L2 regularized case (equation $1$). These symmetries are discrete, while ours are continuous however there is a connection since a mirror symmetry can be obtained from a rescaling symmetry: rescaling a neuron such that its input and output norms are exchanged correspondingly.
> Sparsity is also studied in [2], but it is unstructured sparsity (they count the number of parameter close to zero), while we aim for structured sparsity (disconnecting neurons), and we refer to the related work section of our paper for a few references.
> Finally, [2] studies stability, for example in theorem $1$ and figure $1$, they observe that the mirror surface is stable.
> This has a very clear connection with our framework: this stability can be interpreted as a special case of our invariant set.
> Indeed, the special case of $\mathcal H(0)$ is an invariant set not only for loss only depending on the function, but it also for origin-pointing regularized such losses.
> This is because in $\mathcal{H}(0)$, the regularization gradient has the property of staying in the invariant set.

---

> ### Author Response · Authors · 2025-11-20
>
> - *"It might be useful to also discuss the limitation of studying gradient flow in place of SGD. Because the quantities that obey conservation laws under gradient flow can actually be time-varying in SGD [3,4]."*
>
> It is true that with discrete steps these quantities can vary, however, if we compute the value of $c$ after a single GD or SGD step $\theta_{t+1}=\theta_{t} - h g_{t}$, we see the following:    $$
>     c_{t+1} =B \theta_{t+1}^2 = B(\theta_{t} - h g_{t})^2 = B \theta^2_{t} - 2hB(\theta_{t} \odot g_{t}) + h^2 B g_{t}^2 = c_{t} + h^2 B g_{t}^2.
>     $$
> What this tells us is that, indeed, the value of $c$ drifts due to the finiteness of $h$, but the term responsible for it is *quadratic* in $h$. If $h$ is small and the training time scale is not too large, therefore, we can expect the trajectory to approximately obey the conservation law.
> The experiments on connectedness (see appendix A.9.1 for additional experiments) indicate that the theory approximately holds with discrete steps.
> Nevertheless, we agree that the point you raised still holds and we added a couple of lines in the limitations.
>
> ## Questions
>
> - *"Is there a particular reason to use the uncommon notation of double angle brackets $\langle\langle \rangle\rangle$? I struggled to understand it from a short inline definition given in line 177. I also didn't know if this notation is essential for reading and understanding the main results."*
>
> There is a reason, but we acknowledge it is not evident from the paper.
> The notation is made to remind of an inner product.
> This is because $\langle \langle \theta,g(\theta)\rangle\rangle_{v} := \sum_{i:i\to v} \theta_{(i,v)}g{(i,v)} - \sum_{j:v \to j} \theta_{(v,j)}g{(v,j)} = 0$ is a difference of an inner product over the input (the first sum) and an inner product over the output (the second sum).
> Mathematically, this is usually seen as a bilinear form that makes the parameter space a pseudo-Euclidean space, where 'pseudo' comes from the fact that this bilinear form is not positive definite. Apart from this, the main two reasons we use are to be consistent with previous work on the topic (Nurisso et al. 2024) and to compactly denote the balance equation. We updated the paper by converting the inline equation to a proper equation to improve on readability.

---

> > ### Comment · Reviewer_Fprd · 2025-11-27
> >
> > I thank the authors for their response and helpful revisions. I keep my positive rating of the paper.

---

### Official Review · Reviewer_j6Nf · 2025-10-31

**Soundness:** 3
**Presentation:** 3
**Contribution:** 3
**Rating:** 6
**Confidence:** 3

**Summary:**

This paper studies the properties of the parameter space of ReLU networks, notably in order to decide whether this space is connected and/or contains singularities, which are relevant questions to consider when targeting an optimally trained network, or to prune the network without losing performance/expressiveness, respectively.

The authors consider the framework of Directed Acyclic Graphs (DAGs), which is more general than layered architectures,and focus in this paper on properties of homogeneous activation functions, in particular here the ReLU activation function.

They show a complete characterization of the connectedness of the learning space under GF with any given initialization, analysing to this end the role of bottleneck vertices in the network (that is, vertices with only one out-going arc, or only one in-coming arc) and the balance conditions (invariant under GF once the initialization is done) on related sets of vertices.

Moreover, the authors study singularities, namely parts of the learning space where part of the network stops contributing to the computation. They prove a link between the existence of such singularities and the already mentioned balance conditions, and that even when the conditions are gathered, a GF algorithm will not stumble upon a singularity in finite time.
The authors circumvent this impossibility to favor "self-pruning" by using regularization, and provide numerical experiments showing which regularization helps driving the model towards singularities.

**Strengths:**

Provides a sound and thorough theoretical analysis of the connectivity of learning space for ReLU-activated DAGs Networks trained under GF after arbitrary initialization.

Theoretical analysis of the conditions of existence of singularities, and of the possibility to reach them, complemented with experimental results on tools to reach these singularities in practice.

**Weaknesses:**

The results on connectivity might be achievable with simpler tools and less technicality.

The experimental part on connectivity does not bring anything to the discussion.

The introduction of some notions and symbols is lacking.

**Questions:**

p3, discussing on re-scaling: Do you assume here and in the rest of the paper that all biases are 0?

p4, top of the page: do you have any other requirement on $\\ell$, other than it being differentiable? For instance $\\ell(x,x)=0$ ?

p5, Definition 1: $\\theta^2$ is the vector obtained from $\\theta$ by squaring each individual element? Or do you here implicitly use some other product?

p5, Proposition 3: the point of view of network flows can be obtained in a simpler way as what is done in Appendix A.3. Indeed, since the source and sinks have unconstrained flows, it would suffice to initialize all edge weights with 1, and then correct the balance for each node $u$ with a simple edition of the weights along a path from an arbitrary source to an arbitrary sink going through $u$. Does the algebraic point of view give, in some way, more insight for this paper?

p6, Theorem 1: I think the proof could take a shortcut (following the idea of the precedent remark, and the intuition-providing  text at the beginning of page 7: first prove that if the conditions are not satisfied, it is unfeasible to satisfy the responsible set of vertices, and if it is, show that fixing first the edge weights incident to $Anc(v)$ (or $Desc(v)$) to satisfy the local constraints, and then construct the rest of the solution greedily without editing any edge incident to $Anc(v)$ (which is then possible by definition of this set, anything outside is on at least one path from source to sink avoiding $Anc(v)$).
For the trivial case where the deleted $e$ does not correspond to a bottleneck, the result is immediate.
Then, the Proposition 4 directly yields the result.
Is there a reason for taking the long and more technical way?

p6, Figure 2d: the figure is a good illustration of the proved theorem. I don't understand however, what the additional experiments on real data (Appendix A.9.1) bring to the paper, since it needs no further empirical demonstration that the space is disconnected. As I am less acquainted with the experimental side, could you indicate what I am missing here?

p8, Proposition 6: is the converse known to be true/false?

p9: When and why is self-pruning interesting to have? I understand why one wants to self-prune when the initialization was made such that some singularity exist, but is there an advantage in how expressive the network can be when initialized with a reachable singularity, versus when initialized such that none can be attained by GF?


Typos and Suggestions:

p3, Symmetries of ReLU networks: $\\sigma$ is not introduced, which could be done by adding "the activation function" in front.  Moreover, the formulae for ReLU and Leaky ReLU are both wrong: ReLU: $\\sigma(z)=\\max\\{z, 0\\}$ and Leaky ReLU: $\\sigma(z)=\\max\\{z, \\gamma z\\}$.

p3, Local conservation laws under gradient flow: the variables $d$ and $e$ are clear from context but should be defined nonetheless.

p6, Definition 2: prefer the use of "...with $V^-_B$, $V^+_B$ the sets..., respectively."

p6, Figure 2c: (ii) this case is misleading, since it is not obvious without the text that the case (iii) can "override" it. It also is technically not true, since there could be a completely independent vertex $v'$ in the network making the space disconnected. Maybe find an alternate formulation meaning roughly "a priori connected".

p7, Corollary 2: unless some intermediate layer has a single neuron! It might not be an interesting case, but the soundness of the corollary requires excluding it.

p7: "Concretely, it means that the balance condition ... will forbid sign switches ..." This is the important intuition behind this section, it would be valuable to highlight it more, and potentially to merge it with the previous paragraph.



I am open to updating my rating of the paper, depending on the answers provided by the authors.

---

> ### Author Response · Authors · 2025-11-20
>
> We thank the reviewer for their careful and precise assessment of the results and their proofs.
>
> ## Weaknesses
> - *"The results on connectivity might be achievable with simpler tools and less technicality."*
>
> After reading your comments below, we agree that the manuscript can be simplified and we updated the proof of proposition $3$ to adopt the same construction as in later proofs.
>     For the simplification of the proof of theorem $1$, see our dedicated answer below.
>
> - *"The experimental part on connectivity does not bring anything to the discussion."*
>
> We agree that the theoretical result is complete and, technically speaking, experimentally showing that it holds doesn't add any new insight to it. However, we believe that performing experiments that show its validity in contexts of different complexity, especially due to the fact that real networks cannot be trained with gradient flow (only with gradient descent), is important for clarity purposes. Finally, depending on the reader's background, numerical validation might always be expected.
>
> - *"The introduction of some notions and symbols is lacking."*
>
> We clarified the missing symbols and definitions as suggested in the specific questions made by the reviewer.
>
> ## Questions
>
> 1. Thank you for the opportunity of clarifying this point. As it's standard, one can add biases by adding "virtual" input nodes whose value is fixed to $1$ and connected them to all neurons with bias. The weights of these newly added edges will act as biases. All the results we present in this work already include the bias case if we consider these extra incoming edges to the hidden neurons. We added a sentence explaining this in section $2$, paragraph "DAG neural networks".
> 2. For our results to hold, it is only required that the loss depends on $\theta$ only through the output of the network. This ensures that functionally equivalent parameters have the same loss value.
> 3. Yes, we denote with $\theta^2$ the element-wise square of the vector $\theta$. We added a clarification in Proposition 2.

---

> > ### Author Response · Authors · 2025-11-20
> >
> > 4. Thank you for taking the time to check the proof and for rightly noticing that it can be simplified. We agree that a path-based proof would be simpler to follow and be more in line with the proof of Theorem 1. We updated the proof of Proposition 3 in the supplementary according to your suggestion.
> > 5. We thank you for the insightful comment and for checking the more technical parts of our work. We acknowledge that the proof we provide is quite long and technical but, at the moment, we do not see a straightforward way to simplify it while keeping it rigorous. The steps you propose are indeed intuitive and, in some way, align with our proof. For instance, like you say, when the deleted edge $e$ does not correspond to a bottleneck, one can directly employ Proposition 3, and, in fact, we too use this fact at line 894. However, we do not currently see how to take some specifics aspects of your suggestion and use them to simplify the proof. In particular, two things don't seem trivial to us: 1) proving that if the conditions are not satisfied it is unfeasible to satisfy the responsible set of vertices. 2) Showing that, if the conditions are satisfied, you can fix the edge weights incident to $\mathrm{Anc}(v)$ (or did you mean the pure ancestors?) to satisfy the local constraints. If you could provide more details about what you have in mind, we would be happy to discuss more and improve the proof's readability.
> > 6. We agree that the theorem and the toy example alone are enough to show the presence of disconnectedness and, by themselves, do not need further empirical validation. The real data experiment in Appendix A.9.1 is intended to strengthen the relevance of the result, showing that in the more complex scenario of a larger, more realistic MLP, trained to perform a non-toy task, disconnectedness can still hinder performance, despite the finiteness of the learning rate.
> > 7. Thank you for the interesting question. The converse is not true. Consider, for example, a feed-forward network $G$ made of 4 neurons in a row $\cdot-u-v-\cdot$, with the two hidden neurons $u,v$ initialized with balance values $c_u = 3$ and $c_v = -3$. The neuron set $\{u,v\}$ is such that $c_u + c_v = 0$.  However, $\mathcal{H}_G(c)$ doesn't have singularities as, by Theorem 2, any singular configuration must be of the form $\theta = (0,a,0)$ (which disconnects both $u$ and $v$) or $\theta=(0,0,a)$ (disconnecting $u$) or $\theta=(a,0,0)$ (disconnecting $v$). It is not hard to see that all three types of configuration cannot be in  $\mathcal{H}_G(c)$. In fact, $(0,0,a),(a,0,0)$ will result in having a neuron with balance $0$ (and neither $3$ nor $-3$), while $(0,a,0)$ will give us $c_u = -a^2 \neq 3$ and $c_v = a^2 \neq -3$ for any $a$.
> > 8. With a careful initialization it is possible to include singularities in the invariant set (in particular, this implies respecting the necessary condition discussed above). The advantage in that case would be to allow the training to reduce expressivity along the way, encoding a form of *simplicity bias*, with the caveat of proposition $7$ which says singularities are not reachable in finite time.  From a practical standpoint, a combination of discretization and auxiliary loss allows to overpass proposition $7$. In practice, self-pruning is interesting because it compresses the network, reducing both memory usage and computational cost. In addition, we expect self-pruning to eliminate redundant neurons: cases where a given computation relies on a set of units but could in principle be carried out by a strict subset. This reduction in redundancy can improve the interpretability of the hidden representations, but note that an aggressive compression may also increase superposition (the inverse phenomenon where multiple independent computations may be approximated by fewer units than required to perform them exactly).
> >
> >
> > ## Typos
> >
> > 1. Thank you for spotting the mistake. We corrected it.
> > 2. We added the specification that $f_G(\cdot,\theta):\mathbb{R}^d \to \mathbb{R}^e$ at lin 161.
> > 3. Fixed as suggested.
> > 4. Thank you for correctly pointing this out. We agree that the panel's may be misleading. What we intended to show was that condition (ii) results in the "local" parameter space around $v$ to be connected.We corrected the figure by adding input/output nodes to the condition (ii). Showing a complete network in this way ensures that the invariant set is indeed connected.
> > 5. Thank you for pointing this out. Indeed, if an intermediate layer has a single neuron $v$ the condition of the corollary is not enough to characterize connectedness, as the set of pure descendants and ancestors of $v$ become non-trivial. We adjusted the corollary statement to take this into account.
> > 6. We agree that this intuition is worth putting more emphasis on, and we added this explanation at the end of the paragraph following theorem 1 as suggested.

---

> > > ### Comment · Reviewer_j6Nf · 2025-11-27
> > >
> > > I thank the authors for the detailed answer to my questions, as well as for implementing various fixes to my concerns/suggestions.
> > >
> > > For the point 5., I agree that the points you raise might not be so easy to fix without getting again quite some technicality. And as the current proof works, I am fine with it staying this way.
> > >
> > > I will support acceptance, and update my score accordingly.

---

### Official Review · Reviewer_pzFJ · 2025-11-01

**Soundness:** 4
**Presentation:** 4
**Contribution:** 2
**Rating:** 4
**Confidence:** 4

**Summary:**

The paper studies invariant sets for gradient flow training of DAG-based ReLU architectures and singularities within those invariant sets.

**Strengths:**

The paper is very well written and provides some insights on properties of the training dynamics.

**Weaknesses:**

To me the results seem to be relatively minor and easy extensions of previous results. The authors suggest that formulating these conservation laws with the use of the incidence matrix of the DAG gives significant new insight. But as far as I can see, the main insight is that there a singularities when parts of the graph become disconnected, which does not seem to be surprising.

**Questions:**

On one hand, singularities could be a concern, because they cannot be escaped once reached. On the other hand you suggest that they may be desirable in the sense that they can be seen as the model performing some automatic pruning during training (and indeed you suggest that one may want to induce singularities intentionally). May question is whether there could not be a worry that inducing singularities prematurely limits the model and prevents it from later converging to more favorable solutions which require all neurons (or at least some of the prematurely pruned ones).

---

> ### Author Response · Authors · 2025-11-20
>
> Thank you for your review and for valuing the soundness and clarity of the paper.
> ## Weaknesses
>
> - *"The authors suggest that formulating these conservation laws with the use of the incidence matrix of the DAG gives significant new insight. But as far as I can see, the main insight is that there a singularities when parts of the graph become disconnected"*
>
> First, we would like to clarify that the incidence matrix plays a role not only for the singularities but is a central object in the connectedness analysis.We further argue that the incidence-matrix formulation is not just a change of notation, but what underpins the connectedness analysis in the general feedforward (DAG) setting. Unlike neuron-wise or layer-wise conservation laws used in prior work, this formulation captures how the constraints interact *depthwise* (across layers). This is essential because the connectedness characterization depends on forward/backward stable sets that may span multiple layers. The incidence matrix is precisely the tool that enables this cross-layer coupling to be analyzed systematically.
>
> Regarding the singularity part, we respectfully point out that we go beyond the observation that singularities arise when parts of the graph disconnect. We also provide a dynamical analysis: singularities are stable under GF (proposition $5$), rare (consequence of proposition $6$) and hard to reach (proposition $7$). Finally we operationalize these insights and propose a principled early approach to differentiable structured pruning.
>
>
> - *"easy extensions of previous results."*
>
> We agree that our connectedness analysis extends prior results, but which only applied to shallow architectures. In general feedforward (DAG) networks, the conservation equations become coupled, making the problem significantly more involved.
> In fact, our proof technique, which connects recent results in the topology of intersections of quadrics to network flow theory, is entirely different and could be of independent interest for future theoretical work. Along the way, we also identify a new type of disconnection that does not occur in shallow networks (the plane-cut hyperboloid case in Fig. 2c–d, a consequence of theorem $1$).
>
> Although the pruning literature is large, the singularity part is not a direct extension of an earlier work.
>
> - *"relatively minor results"*
>
> We would like to offer two perspectives on why our results can be worthy of interest.
>
> **Theoretical perspective.**
> In deep-learning settings, our findings are largely *negative* in a useful sense: common architectures (MLPs etc) have many connections and therefore few bottlenecks, and standard initializations (Glorot, Kaiming) tend to assign positive balance values to out-bottlenecks: both strong biases toward connectedness of the optimization space (singularities are also avoided if not aimed for).
> This contributes to understanding why neural networks are typically easy to optimize.
> Moreover, since the invariant set is the space in which all training trajectories reside, we think understanding its properties also bears interest as it constitutes an "intermediate" point of view between the Euclidean structure of the weight space and the, possibly complex, functional space/neuromanifold.
>
> **Practical perspective.**
> Structured pruning seeks efficient models without the hardware inefficiencies of unstructured sparsity. Our results connect this type of pruning to singularities, offering a principled geometric view that may inspire new or more scalable regularization objectives.
> The proposed loss is also quite general: the penalty depends on active neurons regardless of their position, which is a step towards "any-structural pruning" [1].

---

> > ### Author Response · Authors · 2025-11-20
> >
> > ## Questions
> >
> > - *On the early introduction of singularity hurting training.*
> >
> > Thank you for the question which made us reflect on why we did not see this problem empirically.
> >
> > Firstly, inducing singularities limits expressivity so the concern is indeed valid. In the extreme case where the singularity regularizer is weighted much higher than the task loss, the model prunes all neurons and collapses to the zero function. More generally, pruning decreases expressivity; this may help or hurt performance depending on factors like overfitting.
> >
> > Now, more practically, we did not encounter the potential problem you are raising, and we think the reason is linked to the timing of when pruning occurs. In our experiments, neurons are not pruned early in training. For instance, in Fig. 11 (second row), pruning typically begins only around timestep $\sim 1000$ across attributes. As shown in the first row of Fig. 11, this corresponds to a point where the training loss has essentially converged.
> >
> > Our interpretation (which is not a proof) is that early in training, when the task loss still changes, task gradients are dominant compared to the singular loss gradients.  As a result, pruning can only starts after the network has already learned the task. A similar two-phase behaviour has been described for norm-based regularization [2]. For reasonably low values of the singularity-loss weight, training would then unfold in two approximate stages: (1) the model fits the task, and (2) it becomes more efficient by eliminating neurons.
> >
> >
> > - [1] DepGraph: Towards Any Structural Pruning, Fang et al
> > - [2] THE GEOMETRY OF GROKKING: NORM MINIMIZATION ON THE ZERO-LOSS MANIFOLD, Musat et al

---

### Author Response · Authors · 2025-12-01
**Summary of rebuttal phase**

Given the unusual circumstances, we provide a factual summary of the rebuttal contents to help the new AC quickly understand the context:
1. Reviewer pzFJ is the only reviewer who did not respond before the rollback and therefore did not have the chance to revise their assessment based on the discussion.
2. During the discussion, reviewer j6Nf raised their score and supported acceptance.
3. During the discussion, reviewer Fprd confirmed they would maintain their positive score.
4. Reviewer aTae initially posted an incorrect review and quickly updated it after being notified, increasing the score.

Points 1-3 are fully verifiable by looking at the existing discussion. For point 4, the messages titled "Issue with the review" and "Review update" in reviewer aTae's thread as well as the content of the posted review make it clear that the initial review was not the correct one. Point 4 cannot be fully recovered without updating the review of aTae to the state it had after November 13th, which hopefully is still stored on open review. If not, we have our own logs should the AC require them.

---

### Meta-Review · Area_Chair_D1vK · 2026-01-07

**Summary:**

This article studies the invariant parameter space of a ReLU network and associate singularities.

Criticism includes that the results might be incremental and achievable with simpler tools. On the other hand, the article makes valuable observations on the interplay between network topology and optimization geometry, and the overall response has a positive tendency. Therefore I recommend accept.

**Reviewer Concerns:**

Reviewer j6Nf: ``I thank the authors for the detailed answer to my questions, as well as for implementing various fixes to my concerns/suggestions. [...] I will support acceptance, and update my score accordingly.''

Reviewer Fprd: ``I thank the authors for their response and helpful revisions. I keep my positive rating of the paper.''

Reviewer aTae: ``Thanks the authors for the response. I will keep my current score''

**Reviewer Scores:**

For each review, specify how you think the reviewer would have changed their score if they had been able to participate fully in the discussion.

Reviewer pzFJ 4 -> 4
Reviewer j6Nf 6 -> 6
Reviewer Fprd 6 -> 6
Reviewer aTae 6 -> 6

---

### Decision · Program_Chairs · 2026-01-26

Accept (Poster)